# Sub-optimality of the Naive Mean Field approximation for proportional high-dimensional Linear Regression

**Jiaze Qiu**
Department of Statistics
Harvard University
Cambridge, MA 02138, USA
`jiazeqiu@g.harvard.edu`

## Abstract

The Naïve Mean Field (NMF) approximation is widely employed in modern Machine Learning due to the huge computational gains it bestows on the statistician. Despite its popularity in practice, theoretical guarantees for high-dimensional problems are only available under strong structural assumptions (e.g., sparsity). Moreover, existing theory often does not explain empirical observations noted in the existing literature.

In this paper, we take a step towards addressing these problems by deriving sharp asymptotic characterizations for the NMF approximation in high-dimensional linear regression. Our results apply to a wide class of natural priors and allow for model mismatch (i.e., the underlying statistical model can be different from the fitted model). We work under an *iid* Gaussian design and the proportional asymptotic regime, where the number of features and the number of observations grow at a proportional rate. As a consequence of our asymptotic characterization, we establish two concrete corollaries: (a) we establish the inaccuracy of the NMF approximation for the log-normalizing constant in this regime, and (b) we provide theoretical results backing the empirical observation that the NMF approximation can be overconfident in terms of uncertainty quantification.

Our results utilize recent advances in the theory of Gaussian comparison inequalities. To the best of our knowledge, this is the first application of these ideas to the analysis of Bayesian variational inference problems. Our theoretical results are corroborated by numerical experiments. Lastly, we believe our results can be generalized to non-Gaussian designs and provide empirical evidence to support it.

## 1 Introduction

The Naive Mean Field (NMF) approximation is widely employed in modern Machine Learning as an approximation to the actual intractable posterior distribution. The NMF approximation is attractive as (a) it bestows huge computational gains, and (b) it is naturally interpretable and can provide access to easily interpretable summaries of the posterior distribution e.g., credible intervals. However, these two advantages may be overshadowed by the following limitations: (a) it is *a priori* unclear whether this strategy of using a product distribution as a proxy for the true posterior will result in a "good" approximation, and (b) it has been empirically observed that NMF often tends to be significantly over-confident, especially when the feature dimension $p$ is not negligible compared to the sample size $n$. In the traditional asymptotic regime ($p$ fixed and $n \to \infty$), significant progress was made in understanding these two problems for different statistical models, see for instance [8] and references therein. On the other hand, in the complementary high-dimensional regime where both $n$ and $p$ are growing, Ghorbani et al. [7] recently established an instability result for the topic model under the

37th Conference on Neural Information Processing Systems (NeurIPS 2023).

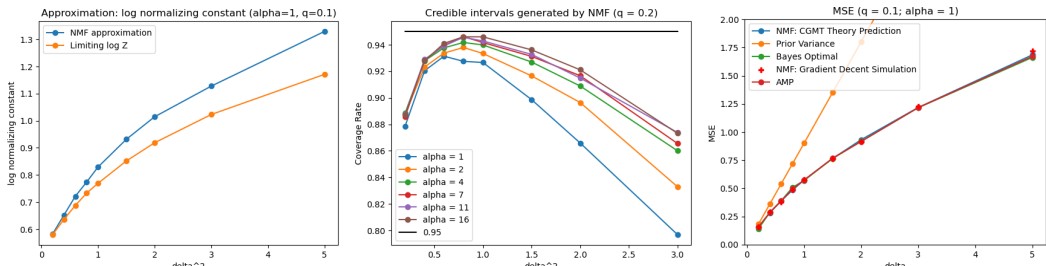

Figure 1: These three figures serve as a visual summary of our main results when the Gaussian Spike and Slab prior is adopted, i.e., NMF does not provide up to leading order correct approximation to the log-normalizing constant (left), and the estimated credible regions suggested by the NMF distribution do not achieve the nominal coverage (middle), even when NMF could achieve close to optimal MSE. Please see Lemma 6 for definitions of the Gaussian Spike and Slab prior and the hyper-parameters $q$ and $\Delta^2$.

proportional asymptotics, i.e., $n = \Theta(p)$. In fact, in this regime, based on non-rigorous physics arguments, it is conjectured and partially established that instead of the NMF free energy one should optimize the TAP free energy. For linear regression in particular, please see [13, 18]. On the other hand, positive results of NMF for high-dimensional linear regression were recently established in [16] when $p = o(n)$.

In this paper, we investigate the performance of NMF approximation for linear regression under the proportional asymptotics regime. As a consequence of our asymptotic characterization, we establish two concrete corollaries: (a) we establish the inaccuracy of the NMF approximation for the log-normalizing constant in this regime, and (b) provide theoretical results backing the empirical observation that NMF can be overconfident in constructing Bayesian credible regions.

Before proceeding further, we formalize the setup under investigation. Given data $\{(y_i, x_i) : 1 \leq i \leq n\}$, $y_i \in \mathbb{R}$, $x_i \in \mathbb{R}^p$, the scientist fits a linear model

$$Y = X\beta^\star + \epsilon, \tag{1}$$

where $\epsilon_i \overset{iid}{\sim} \mathcal{N}(0, \sigma^2)$ and $\beta^\star \in S^p$ is a $p$-dimensional latent signal. We consider either $S = \mathbb{R}$ or $S = [-1, 1]$. In fact, unless explicitly specified otherwise, $S = \mathbb{R}$. Most of our results generalize to bounded support naturally. To recover the latent signal, the scientist adapts a Bayesian approach. She puts an *iid* prior on $\beta_i$'s, namely, $\mathrm{d}\pi_0(\beta) = \prod_{i=1}^p \mathrm{d}\pi(\beta_i)$ and then constructs the posterior distribution of $\beta$,

$$\frac{\mathrm{d}\mu}{\mathrm{d}\pi_0}(\beta) = \frac{\mathrm{d}\mu_{X,y}}{\mathrm{d}\pi_0}(\beta) \propto e^{-\frac{1}{2\sigma^2}\|Y - X\beta\|^2},$$

with normalization constant

$$\mathcal{Z}_p = \mathcal{Z}_p(X, Y) = \int_{S^p} e^{-\frac{1}{2\sigma^2}\|Y - X\beta\|^2} \pi_0(\mathrm{d}\beta). \tag{2}$$

Our results are established assuming that the design matrix $X$ is randomly sampled from an *iid* Gaussian ensemble, i.e., $X_{ij} \overset{iid}{\sim} \mathcal{N}(0, 1/n)$, while we provide empirical evidence for more general classes of $X$ that has *iid* entries with mean 0 and variance $1/n$. Moreover, we assume $n/p \to \alpha \in (0, \infty)$ as $n, p \to \infty$.

**Definition 1** (Exponential tilting). *For any $\gamma := (\gamma_1, \gamma_2) \in \bar{\mathbb{R}} \times \mathbb{R}^+$ and probability distribution $\pi$ on $S$, we define $\pi^\gamma$ as*

$$\frac{\mathrm{d}\pi^\gamma}{\mathrm{d}\pi}(x) := \exp\left(\gamma_1 x - \frac{\gamma_2}{2}x^2 - c(\gamma)\right), \quad c(\gamma) = c_\pi(\gamma) := \log \int_S \exp\left(\gamma_1 x - \frac{\gamma_2}{2}x^2\right)\pi(\mathrm{d}x).$$

*Note that $c(\cdot)$ depends on the distribution $\pi$ and is usually referred to as the cumulant generating function in the statistics literature.*

Using this definition of exponential tilts, the $(X^T X)_{ii}\beta_i^2$ terms in (2) can be absorbed into the base measure

$$\mu(\mathrm{d}\beta) \propto e^{-\frac{1}{2\sigma^2}\|y-X\beta\|^2 + \sum_{i=1}^p \frac{d_i}{2}\beta_i^2} \prod_{i=1}^p \pi_i(\mathrm{d}\beta_i),$$

where $\pi_i := \pi^{(0,d_i)}$ and $d_i := \frac{(X^T X)_{ii}}{\sigma^2}$. By the classical Gibbs variational principle (see for instance [27]), the log-normalizing constant can be expressed as a variational form,

$$-\log \mathcal{Z}_p = \inf_Q \left( \mathbb{E}_Q \left[ \frac{1}{2\sigma^2}\|y - X\beta\|^2 \right] + \mathrm{D}_{KL}\left( Q \| \pi_0 \right) \right)$$

$$= \inf_Q \left( \mathbb{E}_Q \left[ \frac{1}{2\sigma^2}\|y - X\beta\|^2 - \sum_{i=1}^p \frac{d_i}{2}\beta_i^2 \right] + \mathrm{D}_{KL}\left( Q \Big\| \prod_{i=1}^p \pi_i \right) \right) - \sum_{i=1}^p c(0, d_i),$$

where the inf is taken over all probability distribution on $S^p$. While the infimum is always attained if and only if $Q = \mu$, the Naive Mean Field (NMF) approximation restricts the variational domain to product distributions only and renders a natural upper bound,

$$\inf_{Q=\prod_{i=1}^p Q_i} \left[ \mathbb{E}_Q \left( \frac{1}{2\sigma^2}\|y - X\beta\|^2 - \sum_{i=1}^p \frac{d_i}{2}\beta_i^2 \right) + \mathrm{D}_{KL}\left( Q \Big\| \prod_{i=1}^p \pi_i \right) - \sum_{i=1}^p c(0, d_i) \right]. \quad (3)$$

It can be shown that the product distribution $\hat{Q}$ that achieves this infimum is exactly the one closest to $\mu$, in terms of KL-divergence. Before moving forward, we need some additional definitions and basic properties of exponential tilts. The first lemma establishes that instead of using $(\gamma_1, \gamma_2)$ we can also use $(u, \gamma_2) = (\mathbb{E}_{U\sim\pi^\gamma}U, \gamma_2)$ to parameterize the tilted distribution.

**Lemma 1** (Basic properties of the cumulant generating function $c(\cdot)$). *Let $c(\cdot)$ be as in Definition 1. Let $\mathrm{supp}(\pi)$ denote the support of $\pi$. If $m(\pi) := \inf \mathrm{supp}(\pi) < 0$ and $M(\pi) := \sup \mathrm{supp}(\pi) > 0$, then the following conclusions hold.*

    *(a) $\dot{c}(\gamma_1, \gamma_2) := \frac{\partial c(\gamma_1, \gamma_2)}{\partial \gamma_1} = \mathbb{E}_{X\sim\pi^\gamma}(X)$ is strictly increasing in $\gamma_1$, for every $\gamma_2 \in \mathbb{R}$.*

    *(b) For any $u \in (m(\pi), M(\pi))$, there always exists a unique $h(u, \gamma_2) \in \mathbb{R}$ such that $\dot{c}(h(u, \gamma_2), \gamma_2) = u$.*

**Definition 2** (Naive mean field variational objective). *With $d_i := (X^T X)_{ii}/\sigma^2$, we define $M_p(u) : [m(\pi), M(\pi)]^p \to \bar{\mathbb{R}}$ as*

$$M_p(u) := \frac{1}{2\sigma^2}\|y - Xu\|^2 + \sum_{i=1}^p \left[ G(u_i, d_i) - \frac{d_i u_i^2}{2} \right],$$

*where $G$ is defined as a possibly extended real valued function on $[m(\pi), M(\pi)] \times \mathbb{R}$,*

$$G(u, d) := \mathrm{D}_{KL}(\pi^{(h(u,d),d)} \| \pi^{(0,d)}) = uh(u,d) - c(h(u,d),d) + c(0,d) \quad \text{if } u \in (m(\pi), M(\pi)), d \in \mathbb{R},$$

$$:= \mathrm{D}_{KL}\left( \pi_\infty \| \pi^{(0,d)} \right) \quad \text{if } u = M(\pi) < \infty, d \in \mathbb{R},$$

$$:= \mathrm{D}_{KL}\left( \pi_{-\infty} \| \pi^{(0,d)} \right) \quad \text{if } u = m(\pi) > -\infty, d \in \mathbb{R},$$

*in which $h(\cdot, \cdot)$ was defined in Lemma 1 and $\pi_\infty$ and $\pi_{-\infty}$ are degenerate distributions which assigns all measure to $M(\pi)$ and $m(\pi)$ respectively.*

Note that under product distributions, the $\mathbb{E}_Q(\cdot)$ term in (3) is parameterized by the mean vector $u := \mathbb{E}_Q \beta$ and exponential tilts of $\pi_i$'s minimize the KL-divergence term. Therefore, the scaled log-normalizing function, which is also referred to as the average free energy in statistical physics parlance and (log) evidence in Bayesian statistics, is bounded by the following variational form,

$$-\frac{1}{p}\log \mathcal{Z}_p \le \frac{1}{p} \inf_{u\in[m(\pi),M(\pi)]^p} M_p(u) - \frac{1}{p}\sum_{i=1}^p c(0, d_i) = -\frac{1}{p}\log \mathcal{Z}_p^{\mathrm{NMF}}. \quad (4)$$

The right-hand side is equal to (3) and is also referred to as the evidence lower bound (ELBO) or NMF free energy, which can be used as a model selection criterion, see for instance [14]. Asymptotically, the second term is nothing but a constant since it concentrates around $c(0, 1/\sigma^2)$ as $n, p \to \infty$.

The main theoretical question of interest here is whether this bound in (4) is asymptotically tight or not, which serves as the fundamental first step towards answering the question of whether NMF distribution is a good approximation of the target posterior. Please see [4, 27] for comprehensive surveys on variational inference, including but not limited to NMF approximation.

To derive sharp asymptotics for the NMF approximation, the key observation is that the optimization problem is convex under certain priors. We then employ the Convex Gaussian Min-max Theorem (CGMT). CGMT is a generalization of the classical Gordon's Gaussian comparison inequality [9], which allows one to reduce a minimax problem parameterized by a Gaussian process to another (tractable) minimax Gaussian optimization problem. This idea was pioneered by Stojnic [21] and then applied to many different statistical problems, including regularized regression, M-estimation, and so on, see for instance [15, 24]. Unfortunately, concentration results derived from CGMT require both Gaussianity and convexity. This is exactly why we need the Gaussian design assumption in our analysis. In the meantime, though we do not pursue this front theoretically, we provide empirical evidence for more general design matrices in the Supplementary Material. It is worth noting that there is a parallel line of research that aims to develop universality results for these comparison techniques. We refer the interested reader to [11] and references within.

Let us emphasize that our main conceptual concern is not investigating whether (4) as a convex optimization procedure gives a good point estimator, but instead evaluating whether NMF as a strategy or product distributions as a family of distributions can provide "close to correct" approximation for the true posterior. Nevertheless, this optimizer's asymptotic mean square error can also be characterized as a by-product of our main theorem.

Regarding the accuracy of variational approximations in general, certain contraction rates and asymptotic normality results were established in the traditional fixed $p$ large $n$ regime [28, 17, 10]. However, under the high-dimensional setting and scaling we consider in the current paper, without extra structural assumptions (e.g., sparsity), both the true posterior and its variational approximation are not expected to contract towards the true signal, which also explains why one is instead interested in whether the log-normalizing constant can be well approximated, as a weaker standard of "correctness". Ray et al. [19] studied a pre-specified class of mean field approximation in sparse high-dimensional logistic regression. Recently, the first known results on mean and covariance approximation error of Gaussian Variational Inference (GVI) in terms of dimension and sample size were obtained in [12].

## 2 Results

This section starts with some necessary definitions and our main assumptions. Then, we present our main theorem and one natural corollary. Finally, we identify a wide class of priors that would ensure the convexity of the NMF objective, which plays a crucial role in our analysis.

### 2.1 Notations and main assumptions

**Notations:** We use the usual Bachmann-Landau notation $O(\cdot)$, $o(\cdot)$, $\Theta(\cdot)$ for sequences. For a sequence of random variables $\{X_p : p \geq 1\}$, we say that $X_p = o_p(1)$ if $X_p \xrightarrow{P} 0$ as $p \to \infty$ and $X_p = o_p(f(p))$ if $X_p/f(p) = o_p(1)$. We use $C, C_1, C_2 \cdots$ to denote positive constants independent of $n, p$. Further, these constants can change from line to line. For any square symmetric matrix $A$, $\|A\|_{\text{op}}$ and $\|A\|_F$ denote the matrix operator norm and the Frobenius norm, respectively.

**Assumption 1** (Proportional asymptotics). *We assume $n/p \to \alpha \in (0, \infty)$, as $n, p \to \infty$.*

**Assumption 2** (Gaussian features). *For all our theoretical results, we assume the design matrix $X$ is randomly sampled from an iid Gaussian ensemble, i.e., $X_{ij} \overset{iid}{\sim} \mathcal{N}(0, 1/n)$.*

**Definition 3.** *Define $F : (m(\pi), M(\pi)) \to \mathbb{R}$ as*

$$F(u) = F_{\pi, \sigma^2}(u) := G(u, \mathbb{E}d_1) - \frac{u^2 \mathbb{E}d_1}{2} = G\left(u, \frac{1}{\sigma^2}\right) - \frac{u^2}{2\sigma^2}.$$

**Definition 4** (The NMF point estimator). *Recalling the NMF objective $M_p(\cdot)$ as in Definition 2, let $\hat{u} = \hat{\beta}_{NMF} := \arg\min_{u \in [-1,1]^p} M_p(u)$ be the NMF point estimator, which is also the mean vector of the product distribution $(\hat{Q})$ that best approximates the posterior in terms of KL-divergence. We refer to this optimal product distribution as the NMF distribution.*

**Assumption 3** (Convexity of $F(\cdot)$). *We assume $F(\cdot)$ is strongly convex on $S^{\mathrm{o}} := S \setminus \partial S$.*

As alluded to, our analysis relies on the convexity of the "penalty" term $F(\cdot)$. Please note that the definition of $F(\cdot)$ only depends on the prior $\pi$ chosen by the statistician, rather than the "true prior" $\pi^\star$. Therefore, to support this assumption, we provide a few sufficient conditions that identify a broad class of priors that ensure (strong) convexity of $F(\cdot)$ in Section 2.3.

Throughout, we work under a partially well-specified situation, i.e., model (1) is assumed to be correct, but $\beta_i^\star$'s may not have been *a priori* sampled *iid* from $\pi$. Instead, we assume the empirical distribution of $\beta_i^\star$'s converges in $L_2$ to a probability distribution $\pi^\star$ supported on $S$. In addition, the noise level $\sigma^2$ is fixed and known to the statistician. Last but not least, $\pi^\star$ is assumed to have finite second moment and let $s_2 := \mathbb{E}_{T \sim \pi^\star}[T^2] < \infty$.

## 2.2 Main results

From now on, we always assume Assumption 1, 2, and 3. Next, we introduce a scalar denoising function, which is just the proximal operator of $F(\cdot)$.

**Definition 5** (Scalar denoising function). *For $x \in \mathbb{R}$ and $t > 0$,*

$$\eta(x, t) := \arg\min_{w \in S} \left\{ \frac{1}{2t}(w - x)^2 + F(w) \right\} \in S^{\mathrm{o}}$$

Since $F(\cdot)$ is strongly convex, this one-dimensional optimization has a unique minimizer. Note that when $S = [-1, 1]$, since $\lim_{w \to \pm 1} \frac{\mathrm{d}F}{\mathrm{d}w}(w) = \lim_{w \to \pm 1} h(w, 1/\sigma^2) \mp \frac{1}{\sigma^2} = \pm\infty$, the minimum is never achieved on the boundary of $S$. Similarly, when $S = \mathbb{R}$, $\lim_{w \to \pm\infty} \frac{\mathrm{d}F}{\mathrm{d}w}(w) = \pm\infty$. Therefore, the minimum is always achieved at a stationary point. Lastly, $\eta(0, t) = 0$ if $\pi$ is symmetric. In fact, throughout this paper, we only consider symmetric priors.

Before stating our main result and its implications, we first introduce a two-dimensional optimization problem, which will play a central role in our later discussion,

$$\max_{b \geq 0} \min_{\tau \geq \sigma} \phi(b, \tau) \tag{5}$$

$$\phi(b, \tau) := \frac{b}{2}\left(\frac{\sigma^2}{\tau} + \tau\right) - \frac{1}{2}b^2 + \frac{1}{\alpha}\mathbb{E}\min_{w \in S}\left\{ \frac{b}{2\tau}w^2 - bZw + \sigma^2 F(w + B) - \sigma^2 F(B) \right\}, \tag{6}$$

where $F(\cdot)$ was defined in Definition 3 and the $\mathbb{E}$ is taken over $(B, Z) \sim \pi^\star \otimes \mathcal{N}(0, 1)$. In the next lemma, we gather some additional characterizations of this min-max problem.

**Lemma 2.** *The max-min in (5) is achieved at some $(b^\star, \tau^\star) \in (0, \infty) \times (\sigma, \infty)$. In fact, $b^\star$ is unique. In addition, $(b^\star, \tau^\star)$ is also a solution to the following fixed point equation,*

$$\tau^2 = \sigma^2 + \frac{1}{\alpha}\mathbb{E}\left[\left(\eta\left(\tau Z + B, \frac{\tau\sigma^2}{b}\right) - B\right)^2\right]$$

$$b = \tau - \frac{1}{\alpha}\mathbb{E}\left[Z \cdot \eta\left(\tau Z + B, \frac{\tau\sigma^2}{b}\right)\right] = \tau\left(1 - \frac{1}{\alpha}\mathbb{E}\left[\eta'\left(\tau Z + B, \frac{\tau\sigma^2}{b}\right)\right]\right), \tag{7}$$

*where $\eta'(x, t) := \frac{\partial\eta}{\partial x}(x, t)$.*

**Definition 6.** *We use $\nu^\star = \nu_{\pi, \pi^\star}^\star$ to denote the distribution of $\left(\eta\left(\tau^\star Z + B, \frac{\tau^\star\sigma^2}{b^\star}\right), B\right)$, in which $(B, Z) \sim \pi^\star \otimes \mathcal{N}(0, 1)$. We denote by $\hat{\nu}$ the empirical distribution of $\{(\hat{u}_i, \beta_i^\star)\}_{i=1}^p$.*

We are ready to state our main result, which provides a sharp asymptotic characterization of $\hat{\nu}$.

**Theorem 1.** *Suppose the max-min problem in (5) has a unique optimizer $(b^\star, \tau^\star)$, or the fixed point equation in (7) has a unique solution $(b^\star, \tau^\star)$. Then for all $\varepsilon > 0$, as $n, p \to \infty$,*

$$\mathbb{P}\left(W_2\left(\nu^\star, \hat{\nu}\right)^2 \geq \varepsilon\right) \to 0,$$

*where $W_2(\cdot, \cdot)$ stands for order 2 Wasserstein distance.*

**Remark 1.** *This result indicates the NMF estimator $\hat{u}$ should be asymptotically roughly iid among different coordinates, which is different from the NMF distributions being product distributions.*

**Corollary 1.** *Suppose the hidden true signal $\beta^\star$ was a priori sampled iid from a probability distribution $\pi^\star$ with finite second moment. Note that $\pi^\star$ can differ from the prior $\pi$ that the Bayesian statistician chose. In addition, suppose the max-min problem in (5) has a unique optimizer $(b^\star, \tau^\star)$, or the fixed point equation in (7) has a unique solution $(b^\star, \tau^\star)$, then for all $\varepsilon > 0$,*

$$\mathbb{P}\left( W_2 \left( \nu^\star_{\pi, \pi^\star}, \hat{\nu} \right)^2 \geq \varepsilon \right) \to 0, \quad as\ n, p \to \infty,$$

*in which $\nu^\star$ was defined in Definition 6 .*

We provide a proof sketch in Section 6 and all the detailed proofs are deferred to the Supplementary Materials.

## 2.3 Convexity of $F(\cdot)$

In this section, we present a few lemmas that would ensure the validity of Assumption 3. In fact, if conditions of any of these lemmas are satisfied, Assumption 3 holds.

**Lemma 3** (Condition to ensure convexity of $F(\cdot)$: nice prior). *Suppose $\pi$ is absolutely continuous with respect to Lebesgue measure and*

$$\frac{d\pi}{dx}(x) \propto e^{-V(x)}, \forall x \in \text{support}(\pi),$$

*for some $V : \text{support}(\pi) \to \mathbb{R}$. In addition, suppose either of the following two conditions is true,*

1. *$\text{support}(\pi) = \mathbb{R}$; $V(x)$ is continuously differentiable almost everywhere; $V(x)$ is unbounded above at infinity.*

2. *$\text{support}(\pi) = [-a, a]$, for some $0 < a < \infty$; $V(x)$ is continuously differentiable almost everywhere.*

*Then if $V(x)$ is even, non-decreasing in $|x|$ and $V'(x)$ is convex, $F(\cdot)$ is always strongly convex, regardless of the value of $\sigma^2$.*

**Lemma 4** (Condition to ensure convexity of $F(\cdot)$: discrete prior). *Suppose $\pi$ is a symmetric discrete distribution supported on $\{-1, 0, 1\}$,*

$$\pi(dx) = q\delta(x) + \frac{1-q}{2}\delta(x-1) + \frac{1-q}{2}\delta(x+1),$$

*for $q \in (2/3, 1)$. Then $F(\cdot)$ is always strongly convex, regardless of the value of $\sigma^2$.*

Proofs of Lemma 3 and 4 crucially utilize the Griffiths-Hurst-Sherman (GHS) inequality [5, 6], which arose from the study of correlation structure in spin systems. The following two lemmas give examples of some other families of priors for which convexity of $F(\cdot)$ depends on the noise level $\sigma^2$, while those in Lemma 3 and 4 do not.

**Lemma 5** (Condition to ensure convexity of $F(\cdot)$: low signal-to-noise ratio). *Suppose $\text{support}(\pi) \subset [-a, a]$ for some $a > 0$. Then as long as $\sigma^2 > a^2$, $F(u) = F_\pi(u, \sigma^2)$, as a function of $u$, is always strongly convex on S, regardless of the exact choice of $\pi$ and value of $\sigma^2$.*

**Lemma 6** (Condition to ensure convexity of $F(\cdot)$: Spike and Slab prior). *Consider a spike and slab prior to the following form,*

$$\pi(dx) = q\delta(x) + \frac{1-q}{\sqrt{2\pi\Delta^2}} e^{-\frac{x^2}{2\Delta^2}} dx$$

*which is just a mixture of a point mass at 0 and a Normal distribution of mean 0 and variance $\Delta^2$. Suppose*

$$\min_{h \in \mathbb{R}} \text{Var}_{X \sim \pi_{\tilde{q}, \tilde{\Delta}^2}}(X) < \sigma^2 \tag{8}$$

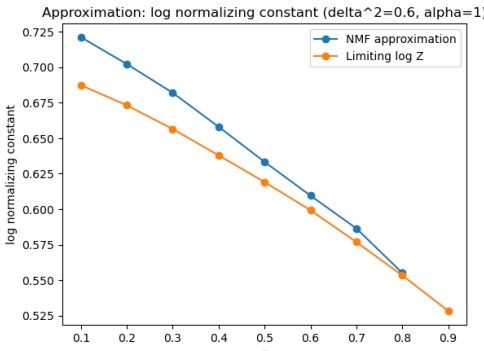
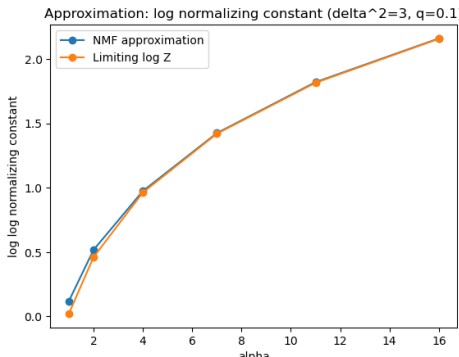

Figure 2: These two figures demonstrate the existence of a gap between $\lim_{p\to\infty}(\mathcal{Z}_p)/p$ and $\lim_{p\to\infty}(\log \mathcal{Z}_p^{\mathrm{NMF}})/p$ when $\pi = \pi^\star$ is a Gaussian Spike and Slab distribution. The left panel features the observation that the gap gets smaller as $q$ (prior sparsity) increases, while the right panel shows as $\alpha := n/p$ gets large, the gap seems to converge to $0$, which is consistent with the results established in [16] when $p = o(n)$.

*where $\pi_{\tilde{q},\tilde{\Delta}^2}$ is again a Gaussian spike and slab mixture,*

$$\pi(\mathrm{d}x) = \tilde{q}\delta(x) + \frac{1-\tilde{q}}{\sqrt{2\pi\tilde{\Delta}^2}}e^{-\frac{x^2}{2\tilde{\Delta}^2}}\,\mathrm{d}x$$

$$with \quad \tilde{q} = \frac{q}{q + (1-q)(1+\Delta^2/\sigma^2)^{-1/2}} \quad and \quad \tilde{\Delta}^2 = \frac{\sigma^2\Delta^2}{\sigma^2+\Delta^2}.$$

*Then, $F(u)$ is strongly convex. In addition, one easier-to-check sufficient condition for* (8) *is*

$$\left(1 + \frac{2q}{1-q}\sqrt{1+\frac{\Delta^2}{\sigma^2}}\right)\frac{\Delta^2}{\sigma^2+\Delta^2} < 1. \tag{9}$$

**Remark 2.** *It is easy to see that for large enough $\sigma$ ($q$ and $\Delta$ fixed), or small enough $q$ ($\Delta$ and $\sigma$ fixed), or small enough $\Delta$ ($q$ and $\sigma$ fixed),* (9) *is always satisfied. In other words, $F(\cdot)$ is strongly convex for low signal-to-noise ratio or high temperature in physics parlance.*

## 3 Log normalizing constant: sub-optimality of NMF

As alluded, as implications of Theorem 1, we develop asymptotics of both $\log \mathcal{Z}_p^{\mathrm{NMF}}$ and mean square error (MSE) of the NMF point estimator $\hat{u}$ in terms of $(b^\star, \tau^\star)$.

**Corollary 2** (MSE). *When conditions of Corollary 1 hold, as $n, p \to \infty$,*

$$\frac{1}{p}\left\|\hat{u} - \beta^\star\right\|^2 \xrightarrow{P} \mathbb{E}_{(B,Z)\sim\pi^\star\otimes\mathcal{N}(0,1)}\left[\left(\eta\left(\tau^\star Z + B, \frac{\tau^\star\sigma^2}{b^\star}\right) - B\right)^2\right] = \alpha(\tau^{\star 2} - \sigma^2).$$

**Corollary 3** (Log normalizing constant). *When conditions of Corollary 1 hold, as $n, p \to \infty$,*

$$-\frac{1}{p}\log \mathcal{Z}_p^{NMF} = \frac{1}{p}\left[M_p(\hat{u}) - \sum_{i=1}^{p}c(0,d_i)\right] \xrightarrow{P} \frac{\alpha b^{\star 2}}{2\sigma^2} + \mathbb{E}F(\eta(B + \tau^\star Z, \tau^\star/b^\star)) - c(0, 1/\sigma^2).$$

Though all our main theorems and corollaries apply to the case when $\pi^\star \neq \pi$, for simplicity and clarity, from now on, we only consider the "nicest" setting, i.e, when assumptions of Corollary 1 are satisfied and in addition $\pi = \pi^\star$. By doing so, we would like to convey that even if there were no model mismatch at all, NMF still would not be "correct".

Concentration and limiting values of both the optimal Bayesian mean square error (i.e., $\mathbb{E}_\mu\|\beta - \mathbb{E}[\beta^\star|X,y]\|^2/p$) and the actual log-normalizing constant were conjectured and rigorously established

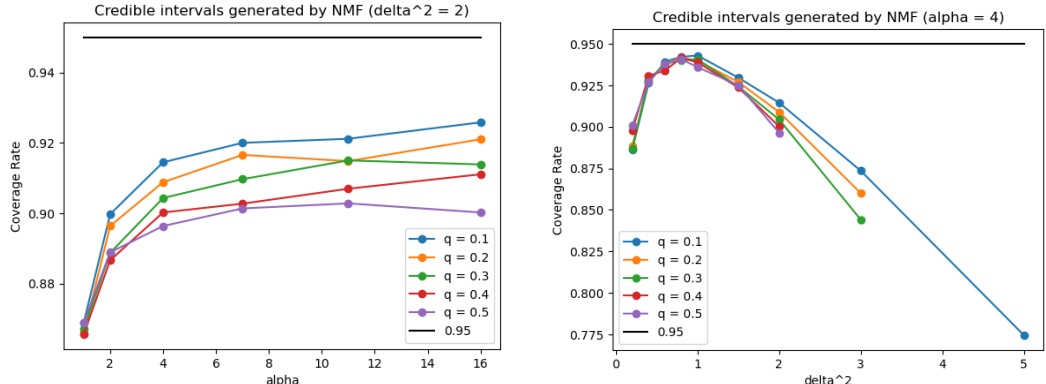

Figure 3: These two figures show that estimated credible regions given by NMF do not achieve the nominal coverage (95%) when $\pi = \pi^\star$ is a Gaussian Spike and Slab distribution. Recall that $\alpha = n/p$ and please see Lemma 6 for exact definitions of the hyper-parameters $q$ and $\Delta^2$.

under additional regularity conditions, which provides us the "correct answers" to compare with. Please see [2, 20].

Please see Figure 2 for numerical evaluations of Corollary 3, which suggests the bound in (4) is not tight for Gaussian Spike and Slab prior. Since, in general, both $F(\cdot)$ and $\eta(\cdot, \cdot)$ lack analytical forms, it is hard to provide a universal guarantee on whether (5) has a unique optimizer or the fixed point equation (7) has a unique solution. In fact, our numerical experiments suggest it is possible for (7) to have multiple fixed points. Therefore, how to exactly realize and evaluate the asymptotic predictions in these two corollaries (so as Corollary 4 in the next section) is challenging in general and can only be done in a case-by-case basis and usually involves numerically solving (7). In light of this observation, we use the Gaussian Spike and Slab prior as defined in Lemma 6 for presentation purposes. Since it is both non-trivial and of practical interest, though, we do emphasize that the same framework and workflow also apply to other priors. Without loss of generality, we also take $\sigma^2 = 1$. This choice renders Figure 2 and 3 in the next section. Details of how to generate these plots are deferred to the Supplementary Material.

## 4    Uncertainty quantification: the average coverage rate

To study uncertainty quantification properties of NMF approximation, we consider the average coverage rate of symmetric Bayesian credible regions (of level $1 - \zeta$) suggested by the NMF distributions, i.e, $R_{p,\zeta} := \frac{1}{p} \sum_{i=1}^{p} \mathbb{1}_{\{\beta_i^\star \in [\hat{q}_{i,\zeta/2}, \hat{q}_{i,1-\zeta/2}]\}}$, where $\hat{q}_{i,t}$ is the $t$-th quantile of $\pi^{(h(\hat{u}_i, d_i), d_i)}$. In order to study asymptotic behavior of $R_{p,\zeta}$, we define an $(m(\pi), M(\pi)) \times S \to \{0, 1\}$ indicator function

$$\psi_\zeta(u_0, \beta_0) = \mathbb{1}_{\left\{ \beta_0 \in \left[ q_{\pi^{(h(u_0, 1/\sigma^2), 1/\sigma^2)}, \zeta/2}, q_{\pi^{(h(u_0, 1/\sigma^2), 1/\sigma^2)}, 1-\zeta/2} \right] \right\}}.$$

The following corollary of Theorem 1 establishes the asymptotic convergence of $R_{p,\zeta}$. Numerically evaluating it for the Gaussian Spike and Slab prior renders Figure 3, which shows NMF credible regions can not achieve the nominal coverage, in this case, 95%, and also provides an exhibition of how large the gaps are for different hyper-parameters.

**Corollary 4.** *Suppose conditions of Corollary 1 hold. In addition, assume the quantile function of $\pi$ is continuous. Then as $n, p \to \infty$,*

$$R_{p,\zeta} \xrightarrow{P} \mathbb{E}_{(B,Z) \sim \pi^\star \otimes \mathcal{N}(0,1)} \left[ \psi_\zeta \left( \eta \left( \tau^\star Z + B, \frac{\tau^\star \sigma^2}{b^\star} \right), B \right) \right].$$

On the other hand, based on the asymptotic joint distribution of $\hat{u}$ and $\beta^\star$ as stated in Corollary1, we can in fact identify a strategy of constructing asymptotically exact Bayesian credible regions based on $\hat{u}$. Let $q_t(x)$ be the $t$-th quantile of conditional distribution of $B$ given $\eta(\tau^\star Z + B, \tau^\star \sigma^2/b^\star) = x$. This way, the following Corollary ensures $[q_{\zeta/2}(\hat{u}_i), q_{1-\zeta/2}(\hat{u}_i)]$ is asymptotically of at least $1 - \zeta$ coverage.

**Corollary 5.** *Suppose conditions of Corollary 4 hold, then for any $\varepsilon > 0$,*

$$\lim_{p \to \infty} \mathbb{P}\left(\frac{1}{p}\sum_{i=1}^{p} \mathbb{1}_{\{\beta_i^\star \in [q_{\zeta/2}(\hat{u}_i), q_{1-\zeta/2}(\hat{u}_i)]\}} < 1 - \zeta - \varepsilon\right) = 0.$$

## 5 Discussion: Extensions and Limitations

In order to provide some intuition on why the NMF approximation is loose in the current setting, it is worth noting that in comparison with the proportional asymptotics regime we consider here, positive results of NMF for high-dimensional linear regression were recently established in [16] when $p = o(n)$. Using terminology from Austin [1], Mukherjee and Sen [16] (when restricted to designs with *iid* Gaussian features) essentially proved, when $p/n \to 0$, the eigenvalue concentration behavior of $X^T X$ leads to the Hamiltonian being of "low complexity". On the other hand, when $p = \Theta(n)$, $\mathrm{tr}(A^2) \neq o(p)$, where $A = A(X)$ is defined as the off-diagonal part of $X^T X$, which violates [16, Equation (5)]. Roughly speaking, when the eigenstructure of A is not "dominated" by a few top eigenvalues, the Hamiltonian can not be covered by an efficient net and thus is not of "low complexity". Please see [1, 16, 3] for more details.

We want to be clear about the fact that, technically, we did not "prove" the sub-optimality of NMF. Instead, we rigorously derived asymptotic characterizations of NMF approximation through the solution of a fixed point equation. But this fixed point equation can only be solved numerically on a case-by-case basis and is not guaranteed a unique solution. All our plots are based on iteratively solving the fixed point equation. As a matter of fact, for instance, when $q$ is close to 1 for the Gaussian Spike and Slab prior we considered, the fixed point equation is clearly not converging to the right fixed point, as demonstrated in the Supplementary Material. It could also just not converge for very small $\alpha$. Nevertheless, all the plots we are showing in the main text are backed by a numerical simulation using simple gradient descent to optimize the NMF objective, i.e. $\inf M_p(u)$, for $n = 8000$. All in all, it is probably more accurate to say we provided a tool for establishing the sub-optimality of NMF for a general class of priors rather than proving it for good.

Another obvious limitation is we can only handle priors that guarantee convexity of the KL-divergence term in terms of the mean parameters. Though it is indeed a broad class of distributions covering some of the most commonly used symmetric priors (e.g., Gaussian, Laplace, and so on), little is known about the asymptotic behavior of NMF when the convexity assumption is violated.

We note that, in theory, in order to carry out the analysis using CGMT, the additive noise $\epsilon$ as defined in (1) does not have to be Gaussian. Instead, as long as it has log-concave density, the same proof idea applies, though we intentionally chose to stick with Gaussian noise as it renders much cleaner results and a more comprehensive presentation. In addition, we expect stronger uniform convergence results (e.g., uniform in $\sigma^2$) could also be established, which can be crucial for applications like hyperparameters selection. Please see [15] for an example in which results of this flavor were obtained.

## 6 Proof strategy

This section gives a proof outline of Theorem 1. More details can be found in the Supplementary Material. Replacing all $d_i$'s in $M_p$ with $\mathbb{E}d_i = 1/\sigma^2$, we define $N_p$ as

$$N_p(u) = \frac{1}{2\sigma^2}\|Y - Xu\|_2^2 + \sum_{i=1}^{p}\left[G(u_i, 1/\sigma^2) - \frac{u_i^2}{2\sigma^2}\right] = \frac{1}{2\sigma^2}\|Y - Xu\|_2^2 + \sum_{i=1}^{p} F(u_i).$$

**Lemma 7.** *Let $\hat{u}_N := \arg\min_u[N_p(u)]$. Then for some $C_s \in \mathbb{R}^+$, as $n, p \to \infty$,*

$$\mathbb{P}\left(\frac{1}{p}\max(\|\hat{u}\|^2, \|\hat{u}_N\|^2) > (1 + C_s)s_2\right) \longrightarrow 0.$$

**Lemma 8.** *For any $\varepsilon > 0$, as $n, p \to \infty$, with $C_s$ as defined in Lemma 7,*

$$\mathbb{P}\left(\frac{1}{p}\sup_{\|u\|^2/p \leq (1+C_s)s_2}\left|\sum_{i=1}^{p}\left[G(u_i, 1/\sigma^2) - \frac{u_i^2}{2\sigma^2}\right] - \left[G(u_i, d_i) - \frac{d_i u_i^2}{2}\right]\right| > \varepsilon\right) \longrightarrow 0. \quad (10)$$

According to Lemma 8 and 7, $N_p(\cdot)$ and $M_p(\cdot)$ are with high probability uniformly close. Thus, from now on, we focus on using Gaussian comparison to analyze $\hat{u}_N$ and $N_p(\hat{u}_N)$ in place of $\hat{u}$ and $M_p(\hat{u})$. Since $F(\cdot)$ is strongly convex, $\hat{w} := \hat{u}_N - \beta^\star$ is the unique minimizer of

$$L(w) := \frac{1}{2n}\|Xw - \epsilon\|^2 + \frac{\sigma^2}{n}\sum_{i=1}^{p}\left(F(w_i + \beta_i^\star) - F(\beta_i^\star)\right).$$

By introducing a dual vector $s$, we get

$$\min_w L(w) = \min_{w \in \mathbb{R}^p} \max_{s \in \mathbb{R}^n} \frac{1}{n}s^T(Xw - \epsilon) - \frac{1}{2n}\|s\|^2 + \frac{\sigma^2}{n}\sum_{i=1}^{p}\left(F(w_i + \beta_i^\star) - F(\beta_i^\star)\right).$$

By CGMT (see for instance [22, Theorem 3.3.1] or [15, Theorem 5.1]), it suffices now to study

$$\min_{w \in \mathbb{R}^p} \max_{s \in \mathbb{R}^n} \frac{1}{n^{3/2}}\|s\|g^T w + \frac{1}{n^{3/2}}\|w\|h^T u - \frac{1}{n}s^T \epsilon - \frac{1}{2n}\|s\|^2 + \frac{\sigma^2}{n}\sum_{i=1}^{p}\left(F(w_i + \beta_i^\star) - F(\beta_i^\star)\right)$$

where $g \sim \mathcal{N}(0, I_p)$ and $h \sim \mathcal{N}(0, I_n)$ and they are independent. Note that the $\min$ and $\max$ can be flipped due to convex-concavity. By optimizing with respect to $s/\|s\|$ and introducing $\sqrt{\frac{\|w\|^2}{n} + \sigma^2} = \min_{\tau \geq \sigma}\left\{\frac{\frac{\|w\|^2}{n} + \sigma^2}{2\tau} + \frac{\tau}{2}\right\}$, it can be further reduced to

$$\max_{b \geq 0} \min_{\tau \geq \sigma} \frac{b}{2}\left(\frac{\sigma^2}{\tau} + \tau\right) - \frac{b^2}{2} + \frac{1}{\alpha}\min_{w \in \mathbb{R}^p}\sum_{i=1}^{p}\left[\frac{1}{p}\left\{\frac{b}{2\tau}w_i^2 - bg_iw_i + \sigma^2 F(w_i + \beta_i^\star) - \sigma^2 F(\beta_i^\star)\right\}\right].$$

Under minor regularity conditions, as $n, p \to \infty$, it converges to

$$\max_{b \geq 0} \min_{\tau \geq \sigma} \frac{b}{2}\left(\frac{\sigma^2}{\tau} + \tau\right) - \frac{b^2}{2} + \frac{1}{\alpha}\mathbb{E}_{B,Z}\min_{w \in \mathbb{R}}\left\{\frac{b}{2\tau}w^2 - bZw + \sigma^2 F(w + B) - \sigma^2 F(B)\right\}$$

with $(B, Z) \sim \pi^\star \otimes \mathcal{N}(0, 1)$, which is how we got $\phi(\cdot, \cdot)$ as in (5). Furthermore, by differentiating $\phi(b, \tau)$ with respect to $\tau$ and $b$, we arrive at the fixed point equation in Lemma 2. Last but not least, note that $\arg\min_w\left\{\frac{b}{2\tau}w^2 - bZw + \sigma^2 F(w + B)\right\} = \eta(\tau Z + B, \tau\sigma^2/b) - B$, which explains why the joint empirical distribution of $(\hat{w}_i, \beta_i^\star)$'s converges to the law of $\left(\eta(\tau^\star Z + B, \tau^\star\sigma^2/b^\star) - B, B\right)$. Finally, we note that similar proof arguments were made in [15, 22].

## Acknowledgements

I am grateful to Subhabrata Sen for some very insightful conversations and encouragement throughout the process. Along with Subhabrata Sen and three anonymous referees, they also made valuable suggestions about earlier versions of this paper. The author was partially supported by a Harvard Dean's Competitive Fund Award to Subhabrata Sen and NSF DMS CAREER 2239234.

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

## Supplementary Materials

## A  Technical lemmas and basic facts

**Lemma 9.** *Let $\dot{c}(h,d) := \frac{\partial c}{\partial h}(h,d)$ and $\ddot{c}(h,d) := \frac{\partial^2 c}{\partial h^2}(h,d)$. We have, for $u \in (m(\pi), M(\pi))$ and $d \in \mathbb{R}$,*

$$\frac{\partial G}{\partial u}(u,d) = h(u,d), \frac{\partial G}{\partial d} = \frac{1}{2}\int_S z^2 \, \mathrm{d}\pi^{(h(u,d),d)}(z) - \frac{1}{2}\int_S z^2 \, \mathrm{d}\pi^{(0,d)}(z).$$

$$\frac{\partial^2 G}{\partial^2 u}(u,d) = \frac{1}{\ddot{c}(h(u,d),d)} = \frac{1}{\mathrm{Var}_{X \sim \pi^{(h(u,d)d)}}(X)} > 0.$$

**Lemma 10** (von Neumann's minimax theorem, [25])**.** *Let $S_w \subset \mathbb{R}^n$ and $S_s \subset \mathbb{R}^m$ be compact convex sets. If $f : S_w \times S_s \to \mathbb{R}$ is a continuous function that is convex-concave, i.e., $f(\cdot, s) : S_w \to \mathbb{R}$ is convex for fixed $s$, and $f(w, \cdot) : S_s \to \mathbb{R}$ is concave for fixed $w$. Then we have that*

$$\min_{w \in S_w} \max_{s \in S_s} f(w,s) = \max_{s \in S_s} \min_{w \in S_w} f(w,s).$$

**Theorem 2** (CGMT, [23, 22, 15])**.** *Let $S_w \subset \mathbb{R}^p$ and $S_s \subset \mathbb{R}^n$ be two compact sets and let $Q : S_w \times S_s \to \mathbb{R}$ be a continuous function. Let $G = (G_{ij})_{1 \le i \le n, 1 \le j \le p} \overset{iid}{\sim} \mathcal{N}(0,1)$, $g \sim \mathcal{N}(0, I_p)$, $h \sim \mathcal{N}(0, I_n)$ be independent standard Gaussian vectors. Denote*

$$\Phi(G) = \min_{w \in S_w} \max_{s \in S_s} s^T G w + Q(w,s),$$

$$\Psi(g,h) = \min_{w \in S_w} \max_{s \in S_s} \|s\|g^T w + \|w\|h^T s + Q(w,s).$$

*Then we have*

1. *For all $t \in \mathbb{R}$,*

$$\mathbb{P}\left(\Phi(G) \le t\right) \le 2\mathbb{P}(\Psi(g,h) \le t).$$

2. *If both $S_w$ and $S_s$ are convex and if $Q(\cdot, \cdot)$ is convex-concave, then for all $t \in \mathbb{R}$,*

$$\mathbb{P}\left(\Phi(G) \ge t\right) \le 2\mathbb{P}(\Psi(g,h) \ge t).$$

**Remark 3.** *The most important message of this theorem is essentially whenever $\Psi(g,h)$ concentrates around a certain value $t$, $\Phi(G)$ will also concentrate around $t$, assuming $Q(\cdot, \cdot)$ is convex-concave.*

## B  Proofs

Proof of Lemma 1 and 9 can be found in for instance [16].

### B.1  Convexity of $F(\cdot)$

*Proof of Lemma 3.* We only prove part (1) here, as proof of part (2) is almost exactly the same. For any $h, d \in \mathbb{R}^+$, by GHS inequality [6, Equation 1.4],

$$\frac{\partial\left[\mathrm{Var}_{B \sim \pi^{(h,d)}}(B)\right]}{\partial h} = \mathbb{E}[B^3] - 3\mathbb{E}B\mathbb{E}[B^2] + 2\left(\mathbb{E}B\right)^3 \overset{\mathrm{GHS}}{\le} 0,$$

Together with the assumption that $V$ is even, we have for any $h \in \mathbb{R}$ and $d \ge 0$,

$$\mathrm{Var}_{B \sim \pi^{(h,d)}}(B) \le \mathrm{Var}_{B \sim \pi^{(0,d)}}(B).$$

Consider now a family of parametric distributions $\{\mathcal{P}_\theta : \theta \ge 0\}$ as a generalization of $\pi^{(0,d)}$, with

$$\frac{\mathrm{d}\mathcal{P}_\theta}{\mathrm{d}x}(x) \propto \exp(-\theta V(x))\exp\left(-dx^2/2\right).$$

Note that $\mathcal{P}_{\theta=1} = \pi^{(0,d)}$. Since $V(\cdot)$ is even and increasing,

$$\mathrm{Var}_{B\sim\pi^{(0,d)}}(B) = \mathrm{Var}_{S\sim\mathcal{P}_{\theta=1}}(S) \leq \mathrm{Var}_{S\sim\mathcal{P}_{\theta=0}}(S)$$

$$= \frac{\int_{\mathbb{R}} z^2 e^{-dz^2/2}\mathrm{d}z}{\int_{\mathbb{R}} e^{-dz^2/2}\mathrm{d}z}$$

$$= \frac{1}{d}\frac{\int_{\mathbb{R}} z^2 e^{-z^2/2}\mathrm{d}z}{\int_{\mathbb{R}} e^{-z^2/2}\mathrm{d}z}$$

$$\leq \frac{1}{d}\,\mathrm{Var}_{S\sim\mathcal{N}(0,1)}(S) = \frac{1}{d},$$

which ensures $\mathrm{Var}_{B\sim\pi^{(h(u,1/\sigma^2),1/\sigma^2)}}(B) \leq \sigma^2$ and therefore $\frac{\mathrm{d}^2 F}{\mathrm{d}u^2}(u) \geq 0$ by (11). Note that as long as $\pi$ is a valid probability distribution, $F(\cdot)$ is not only convex but always strongly convex, as $\mathrm{Var}_{B\sim\pi^{(h(u,1/\sigma^2),1/\sigma^2)}}(B) = \sigma^2$ if and only if $V(\cdot)$ is a constant function and the support of $\pi$ is the whole real line. $\qquad\square$

The same proof idea also applies to Lemma 4; therefore, we omit its proof to avoid redundancy.

*Proof of Lemma 5.* The conclusion follows by noting

$$\frac{\mathrm{d}^2 F}{\mathrm{d}u^2}(u) = \frac{1}{\mathrm{Var}_{B\sim\pi^{(h(u,1/\sigma^2),1/\sigma^2)}}(B)} - \frac{1}{\sigma^2} > 0, \tag{11}$$

as $\pi^{(h,d)}$ is a distribution on $[-a, a]$ and thus its variance is at most $a^2$, which is assumed to be smaller than $\sigma^2$. $\qquad\square$

For Lemma 6, since $\mathrm{Var}_{B\sim\pi^{(h,d)}}(B)$ can be analytically computed for the Gaussian Spike and Slab prior, its proof is nothing but elementary calculation and then checking for (11).

## B.2 Replacing $d_i$ with $\mathbb{E}d_i$

*Proof of Lemma 7.* We focus on only $\|\hat{u}\|$ since almost exactly the same argument also applies to $\hat{u}_N$. We first collect a few high-probability claims, proofs of which are just direct applications of basic standard random matrix results (see, for instance, [26]).

1. There exist positive constants $C_1$ and $C_2$ (only depend on $\alpha$), such that for any $\varepsilon > 0$, $S_1 := \{|\lambda_{\max}(X^T X) - C_1| < \varepsilon\}$ and $S_2 := \{|\lambda_{\min}(X^T X) - C_2| < \varepsilon\}$ are both of high probability.

2. Recall the additive noise $\epsilon \sim \mathcal{N}(0, \sigma^2 I_n)$. For any $\varepsilon > 0$, $S_3 := \{|\|\epsilon\|^2/n - \sigma^2| < \varepsilon\}$ is of high probability.

3. For any $\varepsilon > 0$, $S_4 = \{|\epsilon^T X\beta^\star/p| < \varepsilon\}$ is of high probability.

Let $S_0 = S_1 \cap S_2 \cap S_3 \cap S_4$, which is again an event of approaching 1 probability. Note that since the empirical distribution of $\beta_i^\star$'s converge in $L_2$ to $\pi^\star$, one has $\|\beta^\star\|^2 < 1.01ps_2$ for large enough $p$. When $S_0$ happens, if $\|u\|^2/p > (1 + C_s)s_2$ (with $C_s > 0$ to be chosen later, but large enough such that $\|Xu\| > \|Y\|$),

$$N_p(u) \geq \frac{1}{2\sigma^2}\|Y - Xu\|^2 \geq \frac{1}{2\sigma^2}\left(\|Xu\| - \|Y\|\right)^2 \geq \frac{p}{2\sigma^2}\left[\sqrt{(C_2-\varepsilon)(1+C_s)s_2} - \|X\beta^\star + \epsilon\|/p\right]^2$$

$$\geq \frac{p}{2\sigma^2}\left[\sqrt{(C_2-\varepsilon)(1+C_s)s_2} - \sqrt{2(C_1+\varepsilon)\cdot 2s_2 + 2\alpha(\sigma^2+\epsilon)}\right]^2.$$

On the other hand,

$$N_p\left(\vec{0}\right) = \frac{1}{2\sigma^2}\|Y\|^2 \leq \frac{p}{2\sigma^2}\left[(C_1\varepsilon)\cdot 2ps_2 + \alpha(\sigma^2 - \varepsilon) + 2\varepsilon\right].$$

Upon $C_s$ being large enough, we have $N_p(u) > N_p\left(\vec{0}\right)$ for any $u$ such that $\|u\|^2/p > (1 + C_s)s_2$. Therefore, $\|\hat{u}_N\|^2/p < (1 + C_2)s_2$ on $S_0$. $\qquad\square$

*Proof of Lemma 8.* If $S = [-1, 1]$, by Lemma 9, $\left| \frac{\partial G(u,d)}{\partial d}(u, d) \right| \leq \frac{1}{2}$ for any $u, d$, thus

$$\text{LHS of (10)} \leq \sup_u \left[ \sum_{i=1}^p |G(u_i, d_i) - G(u_i, 1/\sigma^2)| + \sum_{i=1}^p \left| \frac{u_i^2}{2\sigma^2} - \frac{d_i u_i^2}{2} \right| \right]$$

$$\leq \sup_u \left[ \sum_{i=1}^p \left| \frac{\partial G(u,d)}{\partial d}(u_i, 1/\sigma^2)(d_i - 1/\sigma^2) \right| \right] + \frac{1}{2} \sum_{i=1}^p |d_i - 1/\sigma^2|$$

$$\leq \sum_{i=1}^p \left| d_i - \frac{1}{\sigma^2} \right|.$$

Since $X_{ji}$'s are *iid* with variance $1/n$, we know $\mathbb{E}d_i = \frac{1}{\sigma^2} \mathbb{E}\left[ \sum_{j=1}^n X_{ji}^2 \right] = \frac{1}{\sigma^2}$, $d_i \xrightarrow{n\to\infty} \frac{1}{\sigma^2}$, and all $d_i$'s are *iid*, which guarantee RHS of the previous display goes to 0 in probability as $n, p \to \infty$. On the other hand, if $S = \mathbb{R}$, note that for any $\delta \in (0, 1/(2\sigma^2))$, $\mathbb{P}(\max_{1\leq i \leq p} |d_i - 1/\sigma^2| > \delta) \to 0$ as $n, p \to \infty$. In addition, when $\max_{1\leq i \leq p} |d_i - 1/\sigma^2| \leq \delta$ is true, which is of approaching 1 probability,

$$\frac{1}{p} \cdot \text{LHS of (10)} \leq \sup_{u: \|u\|/p < (1+C_s)s_2} \left[ \sum_{i=1}^p |G(u_i, d_i) - G(u_i, 1/\sigma^2)| + \sum_{i=1}^p \left| \frac{u_i^2}{2\sigma^2} - \frac{d_i u_i^2}{2} \right| \right]$$

$$\leq \frac{1}{p} \cdot \sup_{u: \|u\|/p < (1+C_s)s_2} \left[ \sum_{i=1}^p \left| \frac{\partial G(u,d)}{\partial d}(u_i, 1/\sigma^2 + \delta_i)(d_i - 1/\sigma^2) \right| \right] + \frac{1}{2} \sum_{i=1}^p |d_i - 1/\sigma^2| u_i^2,$$

where $\delta_i \in \big( \min(0, d_i - 1/\sigma^2), \max(0, d_i - 1/\sigma^2) \big)$. By Lemma 9, it is further smaller than

$$\frac{1}{p} \cdot \sup_{u: \|u\|/p < (1+C_s)s_2} \left\{ \frac{1}{2} \sum_{i=1}^p \left| d_i - \frac{1}{\sigma^2} \right| \cdot \left[ \text{Var}_{X \sim \pi^{(h(u_i, 1/\sigma^2 + \delta_i), 1/\sigma^2)}}(X) + u_i^2 + \text{Var}_{X \sim \pi^{(0, 1/\sigma^2 + \delta_i)}}(X) + u_i^2 \right] \right\}.$$

Lastly, note that when conditions of one of Lemma 3, 4, 5 and 6 are true, for $\tilde{d}$ close enough to $1/\sigma^2$, we have $\text{Var}_{X \sim \pi^{(\tilde{h}, \tilde{d})}}(X) < 2\sigma^2$ for any $\tilde{h} \in \mathbb{R}$. Therefore, upon choosing small enough $\delta$ such that all $d_i$'s are close enough to $1/\sigma^2$, the display above is controlled by

$$\frac{1}{p} \cdot \sup_{u: \|u\|/p < 2s_2} \left\{ \frac{1}{2} \sum_{i=1}^p \left[ \left| d_i - \frac{1}{\sigma^2} \right| \cdot (4\sigma^2 + 2u_i^2) \right] \right\}$$

$$\leq \max_{1\leq i \leq p} |d_i - 1/\sigma^2| \cdot \sup_{u: \|u\|/p < 2s_2} \left[ 4\sigma^2 + \frac{\|u\|^2}{p} \right]$$

$$\leq \delta \cdot (4\sigma^2 + (1 + C_s)s_2),$$

Lastly, further requiring $\delta < \frac{\varepsilon}{4\sigma^2 + (1 + C_s)s_2}$ renders Lemma 8. $\square$

### B.3 Regarding the fixed point equation

*Proof of Lemma 2.* First of all, recall the definition of $\phi(\cdot, \cdot)$ in (6),

$$\frac{\partial \phi}{\partial b}(b, \tau) = \frac{1}{2}(\sigma^2/\tau + \tau) - b - \frac{\tau}{2\alpha} + \frac{1}{2\tau\alpha} \mathbb{E}\left[ (\tau Z + B - \eta(\tau Z + B, \tau\sigma^2/b))^2 \right].$$

Note that for any fixed $x$, $|x - \eta(x, t)|$ is always strictly increasing with respect to $t$, we have

$$\frac{\partial \left\{ \mathbb{E}\left[ (\tau Z + B - \eta(\tau Z + B, \tau\sigma^2/b))^2 \right] \right\}}{\partial b} < 0,$$

which further leads to

$$\frac{\partial^2 \phi}{\partial b^2}(b, \tau) < -1, \quad \forall b, \tau.$$

Therefore, for any fixed $\tau$, $\phi(\cdot, \tau)$ is 1-strongly concave. Define $\psi(b) := \min_{\tau \geq \sigma} \phi(b, \tau)$. Since $\psi(\cdot)$ is the minimum of a collection of 1-strongly concave functions, it is 1-strongly concave itself and

must have a unique maximizer $b^\star$ over $[0, \infty)$. In addition, by definition of $\eta$, $\lim_{t\to\infty} \eta(x,t) = 0$, dominated convergence theorem gives

$$\lim_{b\to 0^+} \mathbb{E}\left[(\tau Z + B - \eta(\tau Z + B, \tau\sigma^2/b))^2\right] = \mathbb{E}\left[(\tau Z + B)^2\right] = \tau^2 + \mathbb{E}[B^2].$$

Therefore, for any fixed $\tau$,

$$\liminf_{b\to 0} \frac{\partial\phi}{\partial b}(b, \tau) = \frac{1}{2}(\sigma^2/\tau + \tau) + \frac{\mathbb{E}[B^2]}{2\tau\alpha} > 0.$$

Together with Lemma 11 and continuity of $\phi(\cdot, \cdot)$, it ensures $b^\star \neq 0$. On the other hand, for any $b > 0$,

$$\frac{\partial\phi}{\partial\tau}(b, \tau) = \frac{b}{2\tau^2}\left[\tau^2 - \left(\sigma^2 + \frac{1}{\alpha}\mathbb{E}\left[(\eta(\tau Z + B, \tau\sigma^2/b) - B)^2\right]\right)\right],$$

$$\frac{\partial\phi}{\partial\tau}(b, \tau = \sigma) < 0.$$

Together with Lemma 11, we have $\min_{\tau\geq\sigma} \phi(b^\star, \tau)$ has at least one minimizer $\tau^\star \in (\sigma, \infty)$. Finally, since $b^\star$ and $\tau^\star$ are not on the boundary, we have $\frac{\partial\phi}{\partial b}(b^\star, \tau^\star) = \frac{\partial\phi}{\partial\tau}(b^\star, \tau^\star) = 0$, which gives rise to the fixed point equation as in (7). □

**Lemma 11.** *Recall the definition of $\phi$ in (6). For any fixed $b \in (0, \infty)$,*

$$\lim_{\tau\to\infty} \phi(b, \tau) = \infty.$$

*Therefore, $\min_\tau \phi(b, \tau)$ admits at least one minimizer.*

*Proof.* Since $\mathbb{E}[B^2] = s_2 < \infty$, $\mathbb{E}\min_{w\in S}\left\{\frac{b}{2\tau}w^2 - bZw + \sigma^2 F(w + B) - \sigma^2 F(B)\right\}$ is decreasing in $\tau$ and always finite for any $(b, \tau) \in (0, \infty) \times [\sigma, \infty)$. Therefore $\lim_{\tau\to\infty}(b, \tau) = \infty$. □

## B.4 Proof of the main results

We devote this subsection to proving Theorem 1, while we note Corollary 1, 2, 3, 4 and 5 are all direct consequences of it. We first prove Theorem 1 while introducing some necessary lemmas. Then, we prove these lemmas at the end of this subsection. Whenever the optimization domains for $w$ and $s$ are omitted throughout this subsection, they are understood to be $\mathbb{R}^p$ and $\mathbb{R}^n$, respectively. We use $\hat{\nu}$ to denote empirical distribution in general.

Since $F(\cdot)$ is strongly convex, $\hat{w} := \hat{u}_N - \beta^\star$ is the unique minimizer of

$$L(w) := \frac{1}{2n}\|Xw - \epsilon\|^2 + \frac{\sigma^2}{n}\sum_{i=1}^p (F(w_i + \beta_i^\star) - F(\beta_i^\star))$$

By introducing a dual vector $s$, we get

$$\min_w L(w) = \min_{w\in\mathbb{R}^p}\max_{s\in\mathbb{R}^n}\frac{1}{n}s^T(Xw - \epsilon) - \frac{1}{2n}\|s\|^2 + \frac{\sigma^2}{n}\sum_{i=1}^p (F(w_i + \beta_i^\star) - F(\beta_i^\star)) := \min_w\max_s \Phi_X(w, s)$$

Following the recipe in Theorem 2, we define

$$\Psi_{g,h}(w, s) := \frac{1}{n^{3/2}}\|s\|g^T w + \frac{1}{n^{3/2}}\|w\|h^T s - \frac{1}{n}s^T\epsilon - \frac{1}{2n}\|s\|^2 + \frac{\sigma^2}{n}\sum_{i=1}^p (F(w_i + \beta_i^\star) - F(\beta_i^\star)),$$

$$(12)$$

where $g \sim \mathcal{N}(0, I_p)$ and $h \sim \mathcal{N}(0, I_n)$ and they are independent. Note that with a deliberate abuse of notations, we use $\Phi$ and $\Psi$ to denote these two functions to indicate their resemblance to those in the statement of Theorem 2. By Theorem 2, it suffices now to study $\min_w\max_s \Psi_{g,h}(w, s)$ in place of $\min_w\max_s \Phi_X(w, s)$, which is made rigorous by the following lemma.

**Lemma 12.** *Let $D$ be any close set.*

*1. We have for all $t \in \mathbb{R}$*

$$\mathbb{P}\left(\min_{w \in D} \max_s \Phi_X(w, s) \leq t\right) \leq 2\mathbb{P}\left(\min_{w \in D} \max_s \Psi_{g,h}(w, s) \leq t\right).$$

*2. If $D$ is in addition convex, then we have for all $t \in \mathbb{R}$*

$$\mathbb{P}\left(\min_{w \in D} \max_s \Phi_X(w, s) \geq t\right) \leq 2\mathbb{P}\left(\min_{w \in D} \max_s \Psi_{g,h}(w, s) \geq t\right).$$

We defer the proof of Lemma 12 to the end of this section and proceed with proving Theorem 1. Due to strong convexity, $\hat{w}_\Psi := \arg\min_w \max_s \Psi_{g,h}(w, s)$ always exists and is unique. Note that the $\min$ and $\max$ can be flipped due to convex-concavity (Lemma 10). By optimizing with respect to $s/\|s\|$ and introducing

$$\sqrt{\frac{\|w\|^2}{n} + \sigma^2} = \min_{\tau \geq \sigma} \left\{ \frac{\frac{\|w\|^2}{n} + \sigma^2}{2\tau} + \frac{\tau}{2} \right\}, \tag{13}$$

$\min_w \max_s \Psi_{g,h}(w, s)$ can be further reduced to

$$\max_{b \geq 0} \min_{\tau \geq \sigma} \Gamma_{g,h}(b, \tau)$$

$$\Gamma_{g,h}(b, \tau) := \frac{b}{2}\left(\frac{\sigma^2}{\tau} + \tau\right) - \frac{b^2}{2} + \frac{1}{\alpha} \min_{w \in \mathbb{R}^p} \sum_{i=1}^p \left[ \frac{1}{p} \left\{ \frac{b}{2\tau} w_i^2 - b g_i w_i + \sigma^2 F(w_i + \beta_i^\star) - \sigma^2 F(\beta_i^\star) \right\} \right],$$

in the sense that (i) the optimizers $\hat{w}_\Psi$ and $\hat{w}_\Gamma$ are close, i.e, for any $\kappa > 0$,

$$\mathbb{P}\left(\frac{1}{p}\|\hat{w}_\Psi - \hat{w}_\Gamma\|^2 > \kappa\right) \to 0, \tag{14}$$

and (ii) the optimum value is preserved with arbitrarily small error with high probability. The next lemma ensures empirical distribution of $(\hat{w}_\Psi, \beta^\star)$ is close to the distribution of $\left(\eta\left(\tau^\star Z + B, \frac{\tau^\star \sigma^2}{b^\star}\right) - B, B\right)$, which we denote as $\nu^\star_{(w^\star, \pi^\star)}$, where $(B, Z) \sim \pi^\star \otimes \mathcal{N}(0, 1)$.

**Lemma 13.** *Suppose all conditions of Theorem 1 are satisfied. For any $\varepsilon > 0$, there exists $C(\varepsilon) \in (0, \varepsilon)$, such that as $p, n \to \infty$,*

$$\mathbb{P}\left(\exists \tilde{w} \in \mathbb{R}^p \text{ such that } W_2\left(\hat{\nu}_{(\tilde{w}, \beta^\star)}, \nu^\star_{(w^\star, \pi^\star)}\right)^2 \geq \varepsilon \text{ and } \max_s \Psi_{g,h}(\tilde{w}, s) < \min_w \max_s \Psi_{g,h}(w, s) + C(\varepsilon)\right) \to 0.$$

*In the meantime,*

$$\min_w \max_s \Psi_{g,h}(w, s) \xrightarrow{P.} \frac{\alpha b^{\star 2}}{2\sigma^2} + \mathbb{E}F(\eta(B + \tau^\star Z, \tau^\star/b^\star)).$$

Again, for now, we proceed assuming Lemma 13 and prove it later at the end of this section. Building upon Lemma 12 and Lemma 13, we now prove the empirical distribution of $(\hat{u}_N, \beta^\star) = (\beta^\star + \hat{w}, \beta^\star)$ is close to $\nu^\star$ as defined in Definition 6. For $\varepsilon > 0$, define $D_\varepsilon = \left\{ w \in \mathbb{R}^p : W_2\left(\hat{\nu}_{(w, \beta^\star)}, \nu^\star_{(w^\star, \pi^\star)}\right)^2 \geq \varepsilon \right\}$. To establish

$$\mathbb{P}\left(W_2(\hat{\nu}_{(\hat{w}, \beta^\star)}, \nu^\star_{(w^\star, \pi^\star)})^2 > \epsilon\right) \to 0,$$

it suffices to show with high probability, for some $\delta(\varepsilon) > 0$,

$$\min_{w \in D_\varepsilon} \max_s \Phi_X(w, s) \geq \min_{w \in \mathbb{R}^p} \max_s \Phi_X(w, s) + \delta(\varepsilon). \tag{15}$$

On the one hand, by applying both (1) and (2) of Lemma 12 to $D = \mathbb{R}^p$, together with Lemma 13, we have

$$\lim_{n,p \to \infty} \min_w \max_s \Phi_X(w, s) = \lim_{n,p \to \infty} \min_w \max_s \Psi_{g,h}(w, s) = \frac{\alpha b^{\star 2}}{2\sigma^2} + \mathbb{E}F(\eta(B + \tau^\star Z, \tau^\star/b^\star)),$$

where the "lim" is understood to be convergence in probability. It further leads to

$$\mathbb{P}\left(\left|\min_w \max_s \Phi_X(w, s) - \min_w \max_s \Psi_{g,h}(w, s)\right| > \varepsilon\right) \to 0.$$

On the other hand, applying (1) of Lemma 12 to $D = D_\varepsilon$, together with Lemma 13, we have

$$\mathbb{P}\left(\min_{w \in D_\varepsilon} \max_s \Phi_X(w, s) > \min_w \max_s \Phi_X(w, s) + C(\varepsilon) + \varepsilon\right) \to 0,$$

which establishes (15) with $\delta(\varepsilon) = C(\varepsilon) + \varepsilon$, where $C(\varepsilon) > 0$ was defined in Lemma 13. Therefore, we have the empirical distribution of $(\hat{u}_N, \beta^\star)$ is close to the target distribution $\nu^\star$, i.e.,

$$W_2(\hat{\nu}_{(\hat{u}_N, \beta^\star)}, \nu^\star) \xrightarrow{\text{P}} 0. \tag{16}$$

Finally, according to Lemma 8 and 7, $N_p(\cdot)$ and $M_p(\cdot)$ are with high probability uniformly close. Together with strong convexity of $N_p(\cdot)$, we have for any $\kappa > 0$

$$\mathbb{P}\left(\frac{1}{p}\|\hat{u} - \hat{u}_N\|^2 < \kappa\right) \to 0. \tag{17}$$

Theorem 1 is therefore given by (16) and (17). $\qquad\square$

*Proof of Lemma 12.* In order to prove Lemma 12 using Theorem 2, one only needs to establish that the optimizer of $\Phi_X(w, s)$ always has a bounded norm with high probability. In fact, Lemma 7 ensures boundedness of $\hat{w} = \arg\min_w \max_s \Phi_X(w, s)$ while the boundedness of $\hat{s} := \arg\max_s \Phi_X(\hat{w}, s)$ can be established by a similar argument. $\qquad\square$

*Proof of Lemma 13.* Define

$$\tilde{\Gamma}_{g,h}(b, \tau) := \frac{b}{2}\left(\frac{\sigma^2}{\tau} + \tau\right) - \frac{b^2}{2} + \frac{1}{\alpha} \min_{w \in D_\varepsilon} \sum_{i=1}^p \left[\frac{1}{p}\left\{\frac{b}{2\tau}w_i^2 - bg_iw_i + \sigma^2 F(w_i + \beta_i^\star) - \sigma^2 F(\beta_i^\star)\right\}\right].$$

By definition, $\tilde{\Gamma}_{g,h}(b, \tau) \geq \Gamma_{g,h}(b, \tau)$ for any $(b, \tau)$ deterministically. Recall the definition of $\Phi_{g,h}(\cdot)$ in (12),

$$\min_{w \in D_\varepsilon} \max_s \Psi_{g,h}(w, s)$$

$$= \min_{w \in D_\varepsilon} \max_{\|s\|} \frac{1}{n^{3/2}}\|s\|g^T w + \frac{1}{n}\left\|\frac{\|w\|h}{\sqrt{n}} - \epsilon\right\| \cdot \|s\| - \frac{1}{2n}\|s\|^2 + \frac{\sigma^2}{n} \sum_{i=1}^p (F(w_i + \beta_i^*) - F(\beta_i^*))$$

$$\overset{b = \frac{\|s\|}{\sqrt{n}}}{=} \min_{w \in D_\varepsilon} \max_{b \geq 0} \left[\frac{1}{n}bg^T w + b\sqrt{\frac{\|w\|^2}{n} + \sigma^2} - \frac{b^2}{2} + \frac{\sigma^2}{n} \sum_{i=1}^p (F(w_i + \beta_i^*) - F(\beta_i^*))\right] + o_n(1)$$

$$\overset{(13)}{=} \min_{w \in D_\varepsilon} \max_{b \geq 0} \min_{\tau \geq \sigma} \frac{b}{2}\left(\frac{\sigma^2}{\tau} + \tau\right) - \frac{b^2}{2} + \frac{1}{\alpha} \sum_{i=1}^p \left[\frac{1}{p}\left\{\frac{b}{2\tau}w_i^2 - bg_iw_i + \sigma^2 F(w_i + \beta_i^\star) - \sigma^2 F(\beta_i^\star)\right\}\right] + o_n(1)$$

$$\overset{\text{Lemma 10}}{\geq} \max_{b \geq 0} \min_{\tau \geq \sigma} \tilde{\Gamma}_{g,h}(b, \tau) + o_n(1).$$

Therefore,

$$\begin{aligned}
\min_{w \in D_\varepsilon} \max_s \Psi_{g,h}(w, s) &\geq \max_{b \geq 0} \min_{\tau \geq \sigma} \tilde{\Gamma}_{g,h}(b, \tau) + o_n(1) \\
&\geq \min_{\tau \geq \sigma} \tilde{\Gamma}_{g,h}(b^\star, \tau) + o_n(1) \\
&= \tilde{\Gamma}_{g,h}(b^\star, \tilde{\tau}(b^\star)) + o_n(1) \\
&\overset{(i)}{\geq} \Gamma_{g,h}(b^\star, \tilde{\tau}(b^\star)) + o_n(1) \\
&\overset{(ii)}{\geq} \min_{\tau \geq \sigma} \Gamma_{g,h}(b^\star, \tau) + o_n(1) \\
&= \Gamma_{g,h}(b^\star, \tau^\star) + o_n(1) \\
&= \min_{w \in \mathbb{R}^p} \max_s \Psi_{g,h}(w, s) + o_n(1),
\end{aligned} \tag{18}$$

where $\tilde{\tau}(b^\star) := \arg\min_{\tau \geq \sigma} \tilde{\Gamma}_{g,h}(b^\star, \tau)$. We claim that the gaps resulting from (i) and (ii) can not be both negligible. Namely, there exists $\varrho > 0$ such that

$$\limsup_{n,p\to\infty} \mathbb{P}\left(T_1 + T_2 > \varrho\right) > 0, \tag{19}$$

where

$$T_1 := \tilde{\Gamma}_{g,h}(b^\star, \tilde{\tau}(b^\star)) - \Gamma_{g,h}(b^\star, \tilde{\tau}(b^\star)) \geq 0,$$
$$T_2 := \Gamma_{g,h}(b^\star, \tilde{\tau}(b^\star)) - \Gamma_{g,h}(b^\star, \tau^\star) \geq 0.$$

In order to establish (19), we will proceed with proof by contradiction. Suppose (19) is NOT true, equivalently, for any $\varrho > 0$,

$$\lim_{n,p\to\infty} \mathbb{P}\left(T_1 > \varrho\right) = 0, \tag{20}$$
$$\lim_{n,p\to\infty} \mathbb{P}\left(T_2 > \varrho\right) = 0. \tag{21}$$

Recall the definition of $\phi(\cdot)$ in (6). Since $(b^\star, \tau^\star)$ is the unique optimizer of $\phi(\cdot)$ and $\Gamma_{g,h}(b, \tau)$ converges to $\phi(b, \tau)$ uniformly on any compact subset of $[0, \infty) \times [\sigma, \infty)$, (21) is only possible if $|\tilde{\tau}(b^\star) - \tau^*| \overset{\text{P}}{\to} 0$ as $n, p \to \infty$. On the other hand, by definition of $D_\varepsilon$, there exists $\gamma > 0$ such that with high probability, for any $w \in D_\varepsilon$,

$$\sum_{i=1}^{p}\left[\frac{1}{p}\left\{\frac{b^\star}{2\tau^\star}w_i^2 - b^\star g_i w_i + \sigma^2 F(w_i + \beta_i^\star) - \sigma^2 F(\beta_i^\star)\right\}\right]$$
$$> \sum_{i=1}^{p}\left[\frac{1}{p}\left\{\frac{b^\star}{2\tau^\star}\bar{w}_i^2 - b^\star g_i \bar{w}_i + \sigma^2 F(\bar{w}_i + \beta_i^\star) - \sigma^2 F(\beta_i^\star)\right\}\right] + \gamma\varepsilon,$$

where $\bar{w}_i$ is sampled independently from the distribution of $\eta\left(\tau^\star Z + \beta_i^\star, \frac{\tau^\star \sigma^2}{b^\star} - \beta_i^\star\right)$ with $Z \sim \mathcal{N}(0, 1)$. Therefore, with high probability,

$$\tilde{\Gamma}_{g,h}(b^\star, \tau^\star) - \Gamma_{g,h}(b^\star, \tau^\star) > \gamma\varepsilon.$$

By triangle inequality,

$$T_1 \geq \left[\tilde{\Gamma}_{g,h}(b^\star, \tau^\star) - \Gamma_{g,h}(b^\star, \tau^\star)\right] - \left|\tilde{\Gamma}_{g,h}(b^\star, \tilde{\tau}(b^\star)) - \tilde{\Gamma}_{g,h}(b^\star, \tau^\star)\right| - \left|\Gamma_{g,h}(b^\star, \tilde{\tau}(b^\star)) - \Gamma_{g,h}(b^\star, \tau^\star)\right|.$$

In addition, note that if $|\tilde{\tau}(b^\star) - \tau^*| \overset{\text{P}}{\to} 0$ as $n, p \to \infty$, then

$$\tilde{\Gamma}_{g,h}(b^\star, \tilde{\tau}(b^\star)) - \tilde{\Gamma}_{g,h}(b^\star, \tau^\star) \overset{\text{P}}{\to} 0 \quad \text{and} \quad \Gamma_{g,h}(b^\star, \tilde{\tau}(b^\star)) - \Gamma_{g,h}(b^\star, \tau^\star) \overset{\text{P}}{\to} 0.$$

Putting them together, we have

$$\mathbb{P}\left(T_1 > \gamma\varepsilon/2\right) \to 1.$$

The display above is in contradiction to (20), which means (19) is thus established.

Finally, note that Lemma 13 is equivalent to

$$\mathbb{P}\left(\min_{w \in D_\varepsilon} \max_{s} \Psi_{g,h}(w, s) - \min_{w \in \mathbb{R}^p} \max_{s} \Psi_{g,h}(w, s) > C(\varepsilon)\right) \to 1$$

and it is therefore a direct consequence of (19) and (18). $\qquad\qquad\square$

# C  Numerical simulations

All source code can be found in a separate *zip* file in the Supplementary Materials.

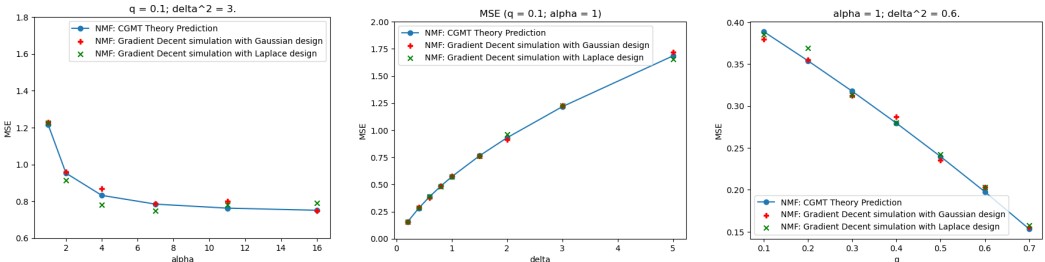

Figure 4: *iid* Gaussian design versus *iid* Laplace design (with Gaussian spike and slab prior): These three plots showcase the empirical observation that prediction of Corollary 2 seem to be valid for a design matrix with *iid* entries that have sub-exponential tails.

## C.1  Universality: non-Gaussian design matrix

Instead of assuming $X_{ij} \overset{iid}{\sim} \mathcal{N}(0, 1/n)$, we now present empirical evidence of universality, i.e., Theorem 1 holds for a broader class of design matrix that has *iid* entries with variance $1/n$. Since it is impossible to exhaust all possible distributions, we will stick with a representative example $X_{ij} \overset{iid}{\sim} \text{Laplace}(\sqrt{2}/2)$ and the Gaussian spike and slab prior. We use Gradient Decent to optimize $M_p(u)$ and then demonstrate that the empirical MSE of its optimizer coincides with the prediction of Corollary 2. Please see Figure 4 for a visual summary.

For more comprehensive and rigorous results on the universality of Gaussian comparison inequalities, we refer interested readers to [11] and references within.

## C.2  Fixed point equation

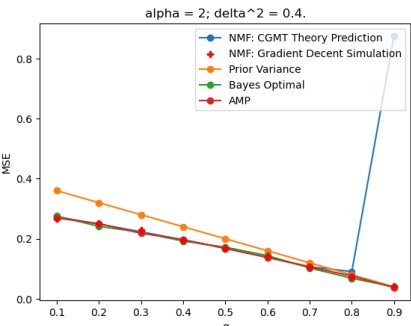

Figure 5: As we can see, when $q$ is large ($q = 0.8$ or $0.9$ in the figure above), our initialization did not lead to the right fixed point.

As alluded to in the main text, all our plots are generated by iteratively solving the fixed point equation (7). However, this naive strategy might not give the right fixed point, i.e., the $(b^\star, \tau^\star)$ that minimizes $\phi(b, \tau)$, or it could just do not converge. Please see Figure 5 for an empirical example. Since either $F(\cdot)$ or $\eta(\cdot, \cdot)$ lacks analytical forms for most natural priors, unlike other applications of CGMT (e.g., asymptotic analysis of lasso [15]), it is hard to determine whether (7) has a unique solution analytically. Fortunately, there are two possible remedies. First, which is the option we took, one could solve $\min_u M_p(u)$ for some large $n$ and check if the empirical MSE matches the prediction given by the fixed point. Alternatively, one could adapt a more brute-force way to find the actual optimizer of $\max_b \min_\tau \phi(b, \tau)$, e.g., grid search or iteratively solving (7) with multiple initializations. After all, it is only a two-dimensional scalar optimization problem. We followed the first way simply because we wanted to use empirical simulations to corroborate our theoretical predictions anyway.

