# OpenReview forum: "Sub-optimality of the Naive Mean Field approximation for proportional high-dimensional Linear Regression"
_NeurIPS.cc/2023/Conference — NeurIPS 2023 poster_

### Official Review · Reviewer_JMXP · 2023-07-03

**Soundness:** 4 excellent
**Presentation:** 2 fair
**Contribution:** 4 excellent
**Rating:** 7
**Confidence:** 3

**Summary:**

This paper derives precise asymptotic characterization for the naive mean field approximation in high-dimensional linear regression under the proportional limit. Two concrete corollaries are obtained including the inaccuracy of the NMF approximation for the log-normalizing constant and the overconfidence of the NMF approximation in terms of uncertainty quantification.

**Strengths:**

1. The naive mean field approximation, the problem studied in the paper, is widely used in modern machine learning. The paper contributes  to understanding the utility of NMF.

2. The paper is mathematically solid and all the technicalities are precisely presented. The proof techniques utilized are quite novel in the area of variational inference.


**Weaknesses:**

I do not find any major weakness of the paper but there are some presentation issues that can be possibly improved. Since the paper is overall very technical and mathematical, I suggest the reviewers to setup a table or a subsection exclusively for all notations. Also, I suggest to considering to formalize conditions utilized throughout the paper such as Gaussian features, noises with special environments of assumptions instead of stating them within texts. The current presentation is a bit challenging for me to follow. Also, there are some typos that need to be fixed. For example, in line 78. I think the definition should be $G(u,d):=D_{KL}(\pi^{(h(u,d),d)}\Vert\pi^{(0,d)})=\cdots$; in line 129, the right most term should be $G(u,\frac{1}{\sigma^2})-\frac{u^2}{2\sigma^2}$.

**Questions:**

1. In line 28, why the map $M_p(u)$ has to be defined over $[-1,1]^p$? It is a bit weird to me because $u=\mathbb{E}[\beta]$ but $\beta$ can be outside the interval.

2. The authors indeed adopt the parametrization based on exponential tilts $\pi^{(h(u,d),d)}$ on the basis of NMF. Does $\pi^{(h(u,d),d)}$ conceptually cover all candidates of continuous distributions? If not, is it the standard setup to facilitate analysis in variational inference?

**Limitations:**

See the weakness stated above.

---

> ### Author Rebuttal · Authors · 2023-08-09
>
> We are grateful for your detailed review of our manuscript and will revise our draft based on your valuable feedback.
>
> Regarding presentation,
>
> 1. To enhance clarity, we will create a subsection (Section 2.1) devoted exclusively to notations and assumptions, such as Gaussian features, which shall be stated in special environments of assumptions instead of within the text. In addition, we will also move Lemma 2-5 to Section 2.3 to avoid burying the main results in these convexity conditions.
>
> 2. We thank the reviewer for pointing out some of the typos. We regret these oversights and will correct them in the revised manuscript.
>
> Answers to questions:
>
> 1. Q: In line 28, why the map $M_p(u)$ has to be defined over $[-1,1]^p$? It is a bit weird to me because $u=\mathbb{E}[\beta]$ but $\beta$ can be outside the interval.
>   - A: The correct domain should be $[m(\pi), M(\pi)]$ instead of $[-1,1]$. We thank the reviewer for bringing this to our attention.
>
> 2. Q: The authors indeed adopt the parametrization based on exponential tilts $\pi^{(h(u, d), d)}$ on the basis of NMF. Does $\pi^{(h(u, d), d)}$ conceptually cover all candidates of continuous distributions? If not, is it the standard setup to facilitate analysis in variational inference?
>   - A: The answer is complex. On the one hand, yes, the exponential tilts are the best choices for minimizing KL divergence between the true posterior and the NMF distribution. On the other hand, if the NMF family was pre-specified, then it might not be covered by the exponential tilts. For example, it is not uncommon for practitioners to fit a Gaussian NMF when the prior is not Gaussian. To summarize, while exponential tilts conceptually may not encompass all candidates of continuous distributions, they include the "best choices." Lastly, we believe it is indeed the standard setup to facilitate analysis in high dimensional variational inference, as illustrated in (Mukherjee and Sen, 2022).

---

> > ### Comment · Reviewer_JMXP · 2023-08-13
> >
> > Thanks for your clarification. I chose to maintain my score of 7.

---

### Official Review · Reviewer_Qw44 · 2023-07-05

**Soundness:** 3 good
**Presentation:** 2 fair
**Contribution:** 3 good
**Rating:** 5
**Confidence:** 3

**Summary:**

The authors provide an asymptotic characterization of naive mean field (NMF) for linear regression under certain priors (e.g., those which guarantee strong convexity).  Under these priors, this characterization can be used to demonstrate the sub-optimality of NMF (asymptotically).

This is primarily a theoretical work, but some experimental observations of the fixed-point scheme are provided.

**Strengths:**

The approach seems novel and fills gaps in the existing literature.

The high-level ideas were easy to follow (though I didn't check all of the proofs in detail).

The authors suggest lots of limitations which may lead to further investigation.

**Weaknesses:**

It isn't immediately clear what practical utility this work would have.  This isn't necessarily required in a theoretical work, but it would bolster the case for acceptance.

Although the main ideas were novel (to me at least), the conditions required to draw the conclusions are difficult to assess (see questions below).

The writing needs a lot of work.  Some notation is undefined (though I could guess what it meant).

**Questions:**

- Could you provide examples of possible practical applications of this work?  The approach seems novel and interesting to me, but I can't ever recall using NMF and thinking that it would yield a highly accurate answer.

- Could you comment on the conditions in Theorem 1 -- are these likely to be true in many instances?  Let's suppose that neither condition is satisfied, is it possible to conclude anything from the fixed point iteration?


Minor typos:
- "i.e." and "e.g." must always be followed by commas.
- "using product" -> "using a product"
- "when feature" -> "when the feature"
-  "progresses were" -> "progress was"
- Stylistically, it is better not to use a citation as a noun.  You should use So and so et al. [citation] instead.
... too many more to log here.  Serious proofreading is recommended, and please, please, please remove the phrase "not gonna be correct."


**Limitations:**

Yes, the authors clearly describe the limitations of the current work, which could suggest interesting avenues for future research.

---

> ### Author Rebuttal · Authors · 2023-08-09
>
> We appreciate the thorough review and constructive suggestions.
>
> We sincerely apologize for language and stylistic issues and will address them in the revised manuscript. We will try our best to learn our lesson and improve our writing skills to mitigate the impacts of language barriers. To further clarify our presentation, we will introduce a subsection (Section 2.1) dedicated to notations and assumptions and relocate Lemma 2-5 to Section 2.3.
>
> Answers to questions:
>
> 1. Q: Could you provide examples of possible practical applications of this work? The approach seems novel and interesting to me, but I can't ever recall using NMF and thinking that it would yield a highly accurate answer.
>   - A: We would like to gently highlight how our work offers practical insights. Generally, in high dimensions, people have recently started understanding that NMF can be very misleading (see, for instance, Ghorbani, Javadi, and Montanari, 2019); on the other hand, other works (e.g., Mukherjee and Sen, 2022) suggest NMF could be accurate to the leading order in some cases depending on the dimensionality. Therefore, there is this tension that we need to understand precisely - when it works and when it does not work at all. Even if it works, how much sub-optimal is it compared to the information-theoretic optimum? Our paper fills in some gaps here and critically explores this model. We agree that we never expect NMF to yield a highly accurate answer in practice. However, our work (1) confirms that NMF is at least not entirely misleading in this setting (high-dimensional linear regression), in contrast with the topic model where NMF is just completely wrong; (2) characterizes the degree to which NMF is sub-optimal, beyond "yes it is sub-optimal." We also hope some of the insights we developed and the technical tools we highlighted in this paper can contribute to analyzing NMF in other more realistic models of higher practical relevance moving forward.
>
> 2. Q: Could you comment on the conditions in Theorem 1 -- are these likely to be true in many instances? Let's suppose that neither condition is satisfied, is it possible to conclude anything from the fixed point iteration?
>   - A: According to physics intuition, as long as one knows what the 'right' fixed point is, then Theorem 1 should still hold. However, we do not have a rigorous proof for this point, so we did not mention it in the manuscript. In addition, we acknowledge the conditions in Theorem 1 are not super straightforward to access. Hopefully, the numerical approach toward the low-dimensional fixed point equation could alleviate it somewhat.

---

### Official Review · Reviewer_6Ztj · 2023-07-06

**Soundness:** 3 good
**Presentation:** 3 good
**Contribution:** 3 good
**Rating:** 7
**Confidence:** 3

**Summary:**

This paper studies the Naive Mean-Field Approximation (NMF) for the posterior distribution of a simple linear regression problem with Gaussian design. It provides an exact asymptotic result for the joint distribution between the NMF approximation $\hat u$ and the regression vector $\beta^\star$. This asymptotic is then used to provide results on the log-partition function associated to the NMF, the mean square error between the NMF estimator and the ground truth, as well as the associated confidence intervals around $\beta^\star$. Those asymptotics depend on a two-dimensional fixed-point equation, which can only be numerically solved; nevertheless, the authors compare this numerical solution with the full posterior distribution, to quantify how the NMF approximates specific quantities of interest of the posterior.

**Strengths:**

This paper is an interesting contribution to the study of Naive Mean-Field estimators. It complements well the results of Sen and Mukherjee on the data-rich setting, by showing that the NMF approximation is not even correct to the first order in the proportional regime. The methods used are fairly standard in the field of exact asymptotics, which is acknowledged in the paper, but they are applied to obtain relevant insights on a new problem. As far as I checked, these methods are applied rigorously and the paper is correct.



**Weaknesses:**

Some of the exposition feels however a bit too obscure: it feels like some very important points (e.g. the definition of the NMF objective and its explanation, or the main theorem) are drowned in very technical considerations (e.g. almost a full page of conditions to ensure the convexity of $F$, or the computation part of the CGMT). I think the paper could achieve a better balance between technical content and explanation, to reach communities that may be familiar with either exact asymptotics or Bayesian approximations, but not both.

Some explanations that would maybe need to be highlighted include:
- why you can replace (3) by the objective in Definition 2, and why you need the $\frac{d_i}2\beta_i^2$ correction term for,
- how the CGMT also imply that the minimizers for both objectives are close (i.e. the part below Lemma 13)

Minor remarks:
- l.38: unfinished sentence ?
- Eq. (3): $\sum_{i=1^p}$
- your log-normalizing constants are sometimes denoted with $Z$ (e.g. eq (2), (4)) and sometimes with $\mathcal{Z}$ (most of the time)
- contrary to what is claimed l.203-207, the expressions for $\mathcal Z_p$ and related quantities are absent from the Supplementary Material
- Eq. (9): I'm not sure redefining $F$ here is necessary
- Fig. 5: there is no right panel

**Questions:**

- How well do you expect the conditions of Theorem 1 to hold ? I read your Appendix C, so I understand why it would be complicated, but do you have at least a few examples of unicity ?
- What's the use of Corollary 1 ? Isn't it strictly weaker than the conditions in l.120-124 ?

---

> ### Author Rebuttal · Authors · 2023-08-09
>
> We thank the reviewer very much for the review.
>
> We appreciate the comments on the general organization of our manuscript. In light of these comments and suggestions from other reviewers, we plan to create a subsection (Section 2.1) devoted exclusively to notations and assumptions, which will be stated in special environments of assumptions instead of being embedded within the text. In addition, we will also move Lemma 2-5 to Section 2.3, hoping to avoid burying the main results among these convexity conditions and appeal to a broader audience.
>
> Replacing (3) with the objective in Definition 2 is made possible by observing that after removing the $d_i \beta_i^2 /2 $ terms, the first term in (3) is a function of $u:= \mathbb{E}_Q \beta$ only and the KL-divergence term in (3) is minimized when $Q_i$ is an exponential tilt of $\pi_i$ (with the constraint that expectation under $Q_i$ is $u_i$).
>
> We concur that the explanations below Lemma 13 can be further improved. The main idea is that Lemma 13, together with applying Lemma 12 part 1 to $ D=D_{\epsilon}$ implies that when $ w $ is within $ D_{\epsilon} $, the optimization can not achieve the optimal value, i.e., the last display on page 16. To further address this concern, we plan to reorganize the part after Lemma 13 (till the end of Appendix B) into formal proof environments instead of floating texts.
>
> Thank you very much for the typos you pointed out in 'minor remarks.' We regret these oversights and will correct them. In particular, a statement of Theorem 3.2 in Barbier et al. (2020) should be added to the appendix. We apologize for the misleading description regarding Fig. 5. There is indeed no right panel there. Instead, we will amend the text in Section C.2 to highlight that "the iterative algorithm might not converge for small $\alpha$."
>
> Answers to questions:
>
> 1. Q: How well do you expect the conditions of Theorem 1 to hold? I read your Appendix C, so I understand why it would be complicated, but do you have at least a few examples of unicity?
>   - A: It is indeed complicated. One example we have in mind is when $\sigma$ is sufficiently large, i.e., the Signal to Noise ratio is very small. This condition essentially corresponds to the ``high temperature'' regime in statistical physics parlance. Though we understand that this setting is not necessarily of great statistical interest, we believe it is nice to have at least some examples.
>
> 2. Q: What's the use of Corollary 1? Isn't it strictly weaker than the conditions in l.120-124?
>   - A: While Corollary 1 is technically weaker than the conditions in L.120-124, we believe it might interest some readers. We emphasize here that our setup indeed covers the most natural well-specified setting.

---

> > ### Comment · Reviewer_6Ztj · 2023-08-15
> >
> > Thank you for your replies ! The outlined changes to the structure of the paper do alleviate my concerns about clarity, and hence I have changed my rating from 6 to 7.

---

### Official Review · Reviewer_zux4 · 2023-07-08

**Soundness:** 3 good
**Presentation:** 3 good
**Contribution:** 3 good
**Rating:** 7
**Confidence:** 4

**Summary:**

This paper studies the naive mean field approximation (NMF) in high dimensional linear regression in the limit where p/n goes to a constant. While the NMF is known to be consistent when n >>p, the authors show that in the high-dimensional case, this is not true. As a result, this paper gives nice theoretic evidence for empirical observations showing that the NMF is no longer accurate in high dimensional regimes.  Moreover, the paper also discusses the consequences on the uncertainty quantification.



**Strengths:**

The research question is important and the results should definitely be published at a top ML conference.

The paper is well-written and easily understandable.  The proof sketch is clear and the theoretical results are nicely illustrated with figures.

**Weaknesses:**

While I recognize that this paper meets the standards for NeurIPS, my reluctance in awarding a higher grade stems from two significant limitations:

1) The proofs' methodological contribution is rather limited. Despite the elegant use of the CGMT, which is commonly seen within the high-dimensional statistics literature, the paper doesn't break new ground in terms of methodology. Although the paper commendably presents the proof in a digestible manner, and the choice of the CGMT is likely the most efficient way to prove the result, this doesn't warrant a significant methodological contribution deserving of a high grade (8+).

2) The insights gained from the results are fairly limited and expected. This is also generally a shortcoming of proofs relying on the CGMT. While such proofs usually yield the precise  asymptotic limit, the resulting theorem statements  are challenging to interpret.


**Questions:**

I am currently missing an intuition in the paper for why the NMF approximation is loose in the high dimensional limit. Can the authors elaborate more on this point?

To me, the main intuition arises from the fact that when $n>>p$, the eigenvalues of $1/n X'X$ concentrate around $1$, which is clearly not the case when $p/n \to c$. As a result, when $n>>p$, the measure $\mu$ is approximately symmetric around its mode (and thus a NMF approximation should be accurate). On the other hand, when $p/n \to c$, the measure $\mu$ is "elliptic" and strongly depends on the small eigenvectors of $X'X$. However, these eigenvectors cannot be captured by the NMF approximation.


**Limitations:**

Yes

---

> ### Author Rebuttal · Authors · 2023-08-09
>
> We thank the reviewer for the thoughtful review and insights. We concur with most of the assessments and find the comments helpful. CGMT as a methodology has indeed been applied in past works, as cited. However, it is noteworthy to highlight that, to the best of our knowledge, it has been exclusively applied to the study of frequentist estimators, while we focus on the possibility of using this machinery to analyze Bayesian methods. We acknowledge the challenges the reviewer pointed out regarding the interpretation of our main theorem, though we hope the numerical approach toward the low-dimensional fixed point equation could alleviate this to some extent.
>
> We concur with the reviewer's intuition regarding why the NMF approximation is loose in the high-dimensional limit. Alternatively, using terminology from (Austin, 2019, "The Structure of Low-Complexity Gibbs Measures on Product Spaces"), when $n/p \to c$, the spectrum behavior of $X^T X$ results in the Hamiltonian no longer being of low complexity, which would further result in the NMF being loose in terms of the log partition function approximation.

---

> > ### Comment · Reviewer_zux4 · 2023-08-10
> > **Response to the rebuttal**
> >
> > I have read the rebuttal and the other reviewers comments.
> >
> > I recommend an acceptance of this paper. However, I strongly encourage the authors to focus more in their paper on the intuition why the NMF approximation is loose. I still think that the CGMT proof is fairly straight forward in this context and well exploited in related (frequentist) settings in the literature. Thus, I would rather put the proof sketch into the appendix and dedicate more space to a better reasoning why this behavior of the NMF is intuitive. Especially given that this is a conference paper, I believe that such a modification could strongly improve the impact of the paper. However, in the end this preference is subjective and may not be shared by other reviewers/the authors.

---

### Author Rebuttal · Authors · 2023-08-09

We thank the Area Chair and the referees for the thoughtful reviews of our paper. We have carefully considered all the comments and will revise the manuscript accordingly.

In light of the reviewers' comments on the general presentation of our manuscript, we plan to:

1. Create a subsection (Section 2.1) devoted exclusively to notations and assumptions, such as Gaussian features, which shall be stated in special environments of assumptions instead of within the text.

2. We will move Lemma 2-5 to Section 2.3 to avoid burying the main results in these convexity conditions.

3. We will reorganize the part after Lemma 13 (till the end of Appendix B) into formal proof environments instead of floating texts to enhance clarity.

Apart from factual questions, which we have responded to in replies to individual reviewers, the reviewers' questions mostly focus on (1) the interpretability of our main theorem and (2) the practical impacts of our work.

Regarding (1), we acknowledge the challenges the reviewers pointed out regarding the interpretation of our main theorem, though we hope the numerical approach toward the low-dimensional fixed point equation could alleviate this to some extent. In addition, since it is hard to give very general conditions for assumptions of Theorem 1 to hold, we would like to advise the practitioners to take a more case-by-case approach, no matter if it is numerical evaluation of the low-dimensional fixed point equation or analytical approaches based on specific forms of the function $F$.

Regarding (2), though we agree that we never expect NMF to yield a "highly" accurate answer in practice, there is this tension that we need to understand precisely - when it works (to some degree) and when it does not work at all. Even if it works, how much sub-optimal is it compared to the information-theoretic optimum? Our paper fills in some gaps here and critically explores this question for the proportional high dimensional linear regression model. For more details, please see our response to Question 1 from Reviewer 3 (Qw44).

Finally, we would like to thank all the reviewers for pointing out some of the typos in our manuscript.

---

### Decision · Program_Chairs · 2023-09-21

**Decision:**

Accept (poster)

**Comment:**

This paper studies proportional asymptotics of the naive mean field approximation in high-dimensional linear models with Gaussian data. Variational inference and high-dimensional asymptotics are certainly key topics of interest in modern statistics and ML. Although their analysis has some clear technical limitations, these are stated explicitly in the work and overall the contribution was appreciated by the reviewers. The authors have also promised some improvements to the exposition after the discussion. Based on this, I recommend acceptance.